

# Inferring the basal sliding coefficient field for the Stokes ice sheet model under rheological uncertainty

Olalekan Babaniyi[1], Ruanui Nicholson[2], Umberto Villa[3], and Noémi Petra[4]

[1]School of Mathematical Sciences, Rochester Institute of Technology, Rochester, NY 14623, USA
[2]Department of Engineering Science, University of Auckland, Auckland 1010, New Zealand
[3]Electrical & Systems Engineering, Washington University in St. Louis, St. Louis, MO 63130, USA
[4]Department of Applied Mathematics, University of California, Merced, Merced, CA 95343, USA

**Correspondence:** Noémi Petra (npetra@ucmerced.edu)

**Abstract.** We consider the problem of inferring the basal sliding coefficient field for an uncertain Stokes ice sheet forward model from surface velocity measurements. The uncertainty in the forward model stems from unknown (or uncertain) auxiliary parameters (e.g., rheology parameters). This inverse problem is posed within the Bayesian framework, which provides a systematic means of quantifying uncertainty in the solution. To account for the associated model uncertainty (error), we employ the Bayesian Approximation Error (BAE) approach to approximately premarginalize simultaneously over both the noise in measurements and uncertainty in the forward model. We also carry out approximative posterior uncertainty quantification based on a linearization of the parameter-to-observable map centered at the maximum a posteriori (MAP) basal sliding coefficient estimate, i.e., by taking the Laplace approximation. The MAP estimate is found by minimizing the negative log posterior using an inexact Newton conjugate gradient method. The gradient and Hessian actions to vectors are efficiently computed using adjoints. Sampling from the approximate covariance is made tractable by invoking a low-rank approximation of the data misfit component of the Hessian. We study the performance of the BAE approach in the context of three numerical examples in two and three dimensions. For each example the basal sliding coefficient field is the parameter of primary interest, which we seek to infer, and the rheology parameters (e.g., the flow rate factor, or the Glen's flow law exponent coefficient field) represent so-called nuisance (secondary uncertain) parameters. Our results indicate that accounting for model uncertainty stemming from the presence of nuisance parameters is crucial. Namely our findings suggest that using nominal values for these parameters, as is often done in practice, without taking into account the resulting modeling error, can lead to overconfident and heavily biased results. We also show that the BAE approach can be used to account for the additional model uncertainty at no additional cost at the online stage.

## 1 Introduction

Inferring the basal sliding coefficient field using both the linear and nonlinear Stokes ice sheet model from noisy surface velocity measurements, has received considerable attention in recent years, see for example Truffer (2004); Raymond and Gudmundsson (2009); Pollard and DeConto (2012); Isaac et al. (2015b); Morlighem et al. (2013); Zhao et al. (2018a, b); Giudici et al. (2014); Petra et al. (2012, 2014); Isaac et al. (2015a). The standard approach to this problem invariably assumes



that the other parameters of the ice, such as those controlling the rheology, are known precisely. This is particularly common, for example, in the case of the so-called flow rate factor and the Glen's flow law exponent, where nominal values such as $A = 10^{-16}\,\mathrm{Pa}^{-n}\,\mathrm{a}^{-1}$ and $n = 3$, respectively, are prescribed; we refer, e.g., to Isaac et al. (2015a); Petra et al. (2014); Raymond and Gudmundsson (2009); Truffer (2004); Morlighem et al. (2013); Zhu et al. (2016); Zhao et al. (2018b); Giudici et al.

(2014); Pollard and DeConto (2012). The inference problem is made significantly more challenging (both theoretically and numerically) by allowing the rheology parameters to be uncertain, and spatially varying. One possible approach to solve the problem is to infer both the basal sliding coefficient and the rheology parameters. However, this considerably increases both the ill-posedness of the inverse problem and the associated computational costs. For most ice sheet inverse problems considered in the literature the field of interest is the basal sliding parameter, which arguably presents the largest uncertainty in determining

the ice flow rate.

It is well documented that, in practice, the rheology parameters of ice sheets are not known exactly (e.g., Bons et al., 2018; Marshall, 2005; Gillet-Chaulet et al., 2011, 2012; Cuffey and Paterson, 2010; Brondex et al., 2019; Raymond and Gudmundsson, 2011). Compounding this issue is the fact that measured ice velocities can be heavily influenced by rheology parameters (Schlegel et al., 2015; Bulthuis et al., 2019). This fact was demonstrated in Petra et al. (2012), where the authors

used the same Stokes ice sheet model as in the current paper to reconstruct reasonable estimates of the Glen's flow law exponent from noisy surface velocity measurements, suggesting that the surface measurements are indeed sensitive to changes in the Glen's flow law exponent field. Despite these findings, it is standard in the literature to assume that rheology—among other—parameters of the ice are known *a priori*, see for instance Bons et al. (2018); Marshall (2005); Gillet-Chaulet et al. (2011, 2012); Cuffey and Paterson (2010); Brondex et al. (2019); Van der Veen (2013).

In this paper, we treat the rheology parameters (specifically the Glen's flow law exponent and the flow rate factor fields) as auxiliary (nuisance) parameters, i.e., parameters which are not of primary interest. However, fixing these auxiliary parameters at incorrect, though possibly well-justified values, often induces so-called *modeling errors*. It is well understood, though, that the solutions to inverse problems are generally sensitive to modeling errors, which—if not properly accounted for—can lead to inaccurate, nonphysical, and in some cases, meaningless solutions of the inverse problem (Brynjarsdóttir and O'Hagan, 2014;

Giudici et al., 2014; Kaipio and Somersalo, 2007, 2005). From a statistical viewpoint, fixing auxiliary parameters to nominal values suggest that these parameters are known exactly, and hence neglects all associated uncertainties. This in turn often results in biased and overconfident estimates for the parameters of interest, see for example Kaipio and Somersalo (2007); Kaipio and Kolehmainen (2013); Nicholson et al. (2018), and the references therein.

We carry out estimation of the basal sliding coefficient within the Bayesian framework (Kaipio and Somersalo, 2005; Stuart,

2010), which is particularly well suited to incorporating various sources and types of uncertainties, including those resulting from model errors (Tarantola, 2005; Kaipio and Somersalo, 2005, 2007). Moreover, to ensure the work here is readily transferable to inference problems in large-scale ice flow problems, such as those discussed in Isaac et al. (2015a), we make use of the computational framework proposed in Bui-Thanh et al. (2013) and Petra et al. (2014) for handling infinite-dimensional Bayesian inverse problems (Stuart, 2010). This approach, combined with adjoint-based means to compute the derivative infor-

mation needed by the optimization solver, ensures mesh independence and computational efficiency.





To account for the uncertainty in the rheology parameters we utilize the Bayesian Approximation Error (BAE) approach (Kaipio and Somersalo, 2005, 2007; Kaipio and Kolehmainen, 2013), which, broadly speaking, lumps all modeling and measurement uncertainties into a single additive *total error* term. The total error can then be approximately marginalized over, in a similar manner to how standard additive errors are dealt with (Kaipio and Kolehmainen, 2013). The BAE approach is particularly attractive computationally as,

(a) the approximate marginalization can be carried out prior to data acquisition, i.e., *premarginalization*, and

(b) the equations to be solved in the adjoint-state approach maintain the same general form (Nicholson et al., 2018).

The BAE approach has been used in a variety of settings, see for example Kaipio and Kolehmainen (2013); Arridge et al. (2006); Castello and Kaipio (2019); Lamien et al. (2019), among others, and the references therein. A particularly relevant, and recent, example is the application of the approach to the so-called *Robin inverse problem* encountered for instance in corrosion detection (Nicholson et al., 2018). There the parameter of interest is also a Robin-type boundary condition on an inaccessible part of the domain, while the nuisance parameter is the (electrical or thermal) conductivity of the domain.

To study the performance of the BAE approach, we formulate and solve three ice sheet flow model problems. Our results suggest that simply setting rheology parameters to nominal values can result in severely misleading estimates of the basal sliding coefficient field, and associated posterior uncertainty, if the additional uncertainty in the rheology parameters is not accounted for. In comparison, we show that incorporating the additional modeling uncertainties using the BAE approach leads to sensible estimates of the basal sliding coefficient and reasonable posterior uncertainty, at no additional online cost. We place particular emphasis on the feasibility of the posterior uncertainty estimates, in particular, on how well the true parameter is contained within the posterior distribution.

**Contributions.** In previous work, we addressed the problem of inferring the basal sliding coefficient field from surface velocity measurements in the context of ice sheet flow in a deterministic, moderate scale, synthetic observational data setting in Petra et al. (2012), in a Bayesian inference and infinite-dimensional setting in Petra et al. (2014), and more recently in a large-scale, real data setting in Isaac et al. (2015a). Here the goal is to extend this inversion framework to account for additional uncertainties in the ice sheet model. The main contributions of this paper are as follows. Firstly we show that setting rheology parameters to values commonly found for ice sheet models in the literature can lead to erroneous posterior estimates of the basal sliding coefficient if the underlying uncertainty in the rheology parameters is not accounted for. Secondly, we show that this situation can be remedied by employing the BAE approach to premarginalize over rheology uncertainties. Thirdly, we show that this approach requires no additional computational resources or time at the *online stage*, as all computations required for premarginalization are carried out prior to the acquisition of data.

**Organization of paper.** The paper is organized as follows. In Section 2, we outline the forward nonlinear Stokes flow equations for ice sheet problems, while in Section 3 we briefly review the Bayesian framework for inverse problems, the computation of the maximum a posteriori estimate and the approximate posterior covariance. In Section 4, we show how to apply the BAE approach to premarginalize over auxiliary parameters. In Sections 5 and 6, we formulate and solve three ice sheet inverse problems and study the performance of the proposed method. Finally, Section 7 provides concluding remarks.





## 2 Forward ice sheet flow model

In this section, we describe the forward ice sheet flow problem that is used for the inference of the basal sliding coefficient field under uncertain rheology. As in Petra et al. (2012, 2014); Isaac et al. (2015a), we model the flow of ice as an isothermal, viscous, shear-thinning, incompressible fluid via the balance of mass and linear momentum (Hutter, 1983; Marshall, 2005;

Paterson, 1994), namely

$$\nabla \cdot \boldsymbol{u} = 0 \qquad \text{in } \Omega, \tag{1a}$$

$$-\nabla \cdot \boldsymbol{\sigma_u} = \rho \boldsymbol{g} \qquad \text{in } \Omega, \tag{1b}$$

where $\boldsymbol{u}$ denotes the velocity field, $\boldsymbol{\sigma_u}$ the stress tensor, $\rho$ the density of the ice, and $\boldsymbol{g}$ gravity. The stress, $\boldsymbol{\sigma_u}$, can be decomposed as $\boldsymbol{\sigma_u} = \boldsymbol{\tau_u} - \boldsymbol{I}p$, where $\boldsymbol{\tau_u}$ is the deviatoric stress tensor, $p$ the pressure, and $\boldsymbol{I}$ the identity tensor. The domain

considered in this paper is $\Omega = [0, L]^{d-1} \times [0, H]$, for $d = 2$ or $d = 3$. We employ the Glen's flow law (Glen, 1955) which relates the stress and strain rate tensors by

$$\boldsymbol{\tau_u} = 2\eta(\boldsymbol{u})\dot{\boldsymbol{\varepsilon}}_{\boldsymbol{u}} \quad \text{with} \quad \eta(\boldsymbol{u}) = \frac{1}{2} A^{-\frac{1}{n}} \dot{\varepsilon}_{\text{II}}^{\frac{1-n}{2n}}, \tag{1c}$$

where $\eta$ is the effective viscosity, $A$ is the flow rate factor, $\dot{\boldsymbol{\varepsilon}}_{\boldsymbol{u}} = \frac{1}{2}(\nabla \boldsymbol{u} + \nabla \boldsymbol{u}^T)$ and $\dot{\varepsilon}_{\text{II}} = \frac{1}{2}\text{tr}(\dot{\boldsymbol{\varepsilon}}_{\boldsymbol{u}}^2)$ are the strain rate tensor and its second invariant. Above, $n = n(\boldsymbol{x})$ is the spatially varying Glen's flow law exponent, which satisfies $n(\boldsymbol{x}) \geq 1$ for all

$\boldsymbol{x} \in \Omega$ to ensure the ice is a shear-thinning fluid (Glen, 1955).

Inline with Petra et al. (2012), the top boundary $\Gamma_\text{t}$ is equipped with a traction-free boundary condition, all lateral boundaries $\Gamma_\text{p}$ are equipped with periodic boundary conditions, and on the basal surface $\Gamma_\text{b}$ we apply a no flow condition for the normal component of $\boldsymbol{u}$ along with a linear sliding law for the tangential components. That is, the boundary conditions are given by

$$\boldsymbol{\sigma_u}\boldsymbol{n} = \boldsymbol{0} \qquad \text{on } \Gamma_\text{t}, \tag{1d}$$

$$\boldsymbol{u}|_{\Gamma_\text{l}} = \boldsymbol{u}|_{\Gamma_\text{r}} \text{ and } \boldsymbol{\sigma_u}\boldsymbol{n}|_{\Gamma_\text{l}} = \boldsymbol{\sigma_u}\boldsymbol{n}|_{\Gamma_\text{r}} \qquad \text{on } \Gamma_\text{p}, \tag{1e}$$

$$\boldsymbol{u} \cdot \boldsymbol{n} = 0 \qquad \text{on } \Gamma_\text{b}, \tag{1f}$$

$$\boldsymbol{T}\boldsymbol{\sigma_u}\boldsymbol{n} + \exp(\beta)\boldsymbol{T}\boldsymbol{u} = \boldsymbol{0} \qquad \text{on } \Gamma_\text{b}, \tag{1g}$$

where $\beta(\boldsymbol{x})$ is the log basal sliding coefficient field[1], $\boldsymbol{n}$ is the outward normal unit vector, and $\boldsymbol{T} := \boldsymbol{I} - \boldsymbol{n}\boldsymbol{n}^T$ is the projection onto the tangential plane. Above we generically used $\Gamma_\text{l}$ and $\Gamma_\text{r}$ to denote pairs of opposing boundaries on $\Gamma_\text{p}$ on which periodic

conditions are imposed. We note that $\beta$ generally represents a combination of complex phenomena, see for example Schoof (2005, 2010); Perego et al. (2014). Furthermore, the methods and results discussed in the current paper do not rely on the particular top and lateral boundary conditions specified. As such, alternative boundary conditions could also be imposed on $\Gamma_\text{t}$ and $\Gamma_\text{p}$. For a simple illustration of the problem set up (shown in two dimensions) see Fig. 1.

---

[1]The 'exp' function is used to ensure the basal sliding coefficient remains positive. For simplicity, in what follows, we will refer to $\beta$ as the basal sliding coefficient.





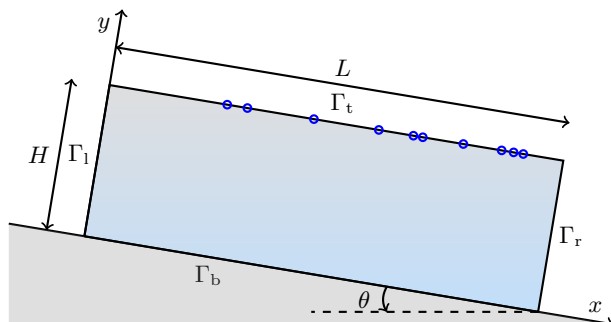

**Figure 1.** Schematic of a two-dimensional slab of ice (used in Examples 1 and 2). The schematic can also be thought of as a cross section through the three-dimensional slab of ice used for Example 3. The blue circles show representative (random) measurement locations, but do not necessarily coincide with the actual measurement locations used in the examples. $\theta$ is the slope of the ice slab.

*The weak form of the Stokes equation.* In what follows, let us introduce the weak form of (1), as it is the starting point for both the finite element discretization of the forward model and the computation of the gradient and action of the Hessian required for the solution of the inverse problem using the adjoint-state method, see e.g., Isaac et al. (2015a). Multiplying the nonlinear Stokes system (1) with arbitrary tests functions $\tilde{u}$ and $\tilde{p}$ and using integration by parts over $\Omega$ (Gockenbach, 2006;

Elman et al., 2005), the weak form of (1) is given by: Find $(u, p) \in \mathcal{W} = \mathcal{V} \times \mathcal{Q}$ such that

$$
\int_{\Omega} 2\eta(u)\dot{\varepsilon}_u : \dot{\varepsilon}_{\tilde{u}} \, dx - \int_{\Omega} (p\nabla \cdot \tilde{u} + \tilde{p}\nabla \cdot u) \, dx
$$
$$
+ \int_{\Gamma_b} \exp(\beta)Tu \cdot T\tilde{u} \, ds = \int_{\Omega} \rho g \cdot \tilde{u} \, dx, \tag{2}
$$

for all $(\tilde{u}, \tilde{p}) \in \mathcal{W}$. Inline with Elman et al. (2005); Isaac et al. (2015b), we set $\mathcal{V} := \{u \in (H^1(\Omega))^d : u|_{\Gamma_l} = u|_{\Gamma_r}, \, u \cdot n|_{\Gamma_b} = 0\}$ and $\mathcal{Q} := (L^2(\Omega))^d$, for $d = 2$ or $d = 3$.

*Discretization.* To guarantee the *inf-sup* stability (well-posedness) of the discretized forward problem, we discretize the

velocity and pressure using Taylor-Hood finite elements, i.e., quadratic elements for each velocity component, and linear elements for pressure, see for example Elman et al. (2005). The basal sliding coefficient field is discretized using continuous linear Lagrange basis functions $\{\phi_j(s)\}_{j=1}^m$, i.e., $\beta_h(s) = \sum_{j=1}^m \beta_j \phi_j(s)$, where $s \in \Gamma_b$. In what follows, we denote by $\beta = (\beta_1, \beta_2, \ldots, \beta_m) \in \mathbb{R}^m$ the discrete basal sliding coefficient field.

## 3   Inferring the basal sliding coefficient field

In this section, we summarize the Bayesian inference framework, which will be used in combination with the Bayesian approximation error approach, to account for uncertainties in rheology parameters. To allow for systematic incorporation of uncertainties, we consider the inverse problem in the Bayesian framework (Tarantola, 2005; Kaipio and Somersalo, 2005). In this framework, the solution of the underlying statistical inverse problem is given by the posterior probability density. For non-





linear inverse problems with expensive forward models and high-dimensional parameters (as is the case for ice sheet inverse problems), fully characterizing the posterior is typically not tractable. Consequently, we compute the Laplace approximation of the posterior, which requires only the *maximum a posteriori* (MAP) estimate, i.e., the basal sliding coefficient which maximizes the posterior density and the approximate posterior covariance.

We use Bayes' Theorem to write the solution of the Bayesian inverse problem as the posterior measure, which describes the probability law of the parameter conditioned on measurements (Tarantola, 2005; Stuart, 2010). Formulation of the posterior relies on both the prior density and the likelihood function, which we outline below. We note that, initially we pose the problem in an infinite-dimensional setting, which is particularly well suited to large-scale problems (e.g., Bui-Thanh et al., 2012; Isaac et al., 2015a), as it ensures discretization invariance and well-posedness of the Bayesian inverse problem (Stuart, 2010).

**3.1   The prior**

We postulate a Gaussian prior density on the (spatially varying) basal sliding coefficient, i.e., $\beta \sim \mathcal{N}(\beta_*, \mathcal{C}_\beta)$, with covariance operator $\mathcal{C}_\beta$, and mean value $\beta_* \in \mathcal{E}$ where $\mathcal{E}$ is defined as the range of $\mathcal{C}_\beta^{\frac{1}{2}}$, see for example Stuart (2010); Bui-Thanh et al. (2013) for more details. To ensure the inverse problem is well-posed in infinite dimensions, we use a squared inverse elliptic operator to define the prior covariance operator (e.g., Flath et al., 2011; Bui-Thanh et al., 2013; Petra et al., 2014).

More specifically, we take $\mathcal{C}_\beta = \mathcal{A}^{-2}$, where $\mathcal{A}$ is the second order elliptic differential operator defined by

$$\mathcal{A}\beta := -\nabla \cdot (\gamma_\beta \nabla \beta) + \delta_\beta \beta \quad \text{on } \Gamma_b, \tag{3}$$

where parameters $\gamma_\beta > 0$ and $\delta_\beta > 0$ control the correlation length and the marginal variance. As discussed in Khristenko et al. (2019); Daon and Stadler (2018); Roininen et al. (2014), suitable boundary conditions need to be stipulated to reduce boundary artifacts. In this work we choose to equip $\mathcal{A}$ with periodic boundary conditions on $\partial\Gamma_b$, which parallels the periodic boundary
condition (1e) of the forward model. We note that the discrete representation of the prior covariance operator, denoted $\mathbf{\Gamma}_{\text{pr}}$, is defined as (e.g., Bui-Thanh et al., 2013; Petra et al., 2014; Villa et al., 2020)

$$\left[\mathbf{\Gamma}_{\text{pr}}^{-1}\right]_{ij} = \int_{\Gamma_b} \phi_i(\boldsymbol{s}) \mathcal{A}^2 \phi_j(\boldsymbol{s}) d\boldsymbol{s} \quad i,j \in \{1, 2, \ldots, m\}. \tag{4}$$

Therefore, the discrete parameter $\boldsymbol{\beta}$ follows a Gaussian distribution $\mathcal{N}(\boldsymbol{\beta}_*, \mathbf{\Gamma}_{\text{pr}})$, with prior mean $\boldsymbol{\beta}_* \in \mathbb{R}^m$ and covariance $\mathbf{\Gamma}_{\text{pr}}$. That is the prior density of $\boldsymbol{\beta}$ is given by

$$\pi_{\text{prior}}(\boldsymbol{\beta}) \propto \exp\left\{-\frac{1}{2} \|\boldsymbol{\beta} - \boldsymbol{\beta}_*\|_{\mathbf{\Gamma}_{\text{pr}}^{-1}}^2\right\}, \tag{5}$$

where $\|\cdot\|_{\mathbf{\Gamma}_{\text{pr}}^{-1}}$ denotes the $\mathbf{\Gamma}_{\text{pr}}^{-1}$ weighted $l_2$ inner product.





### 3.2 The data likelihood

We assume the velocity measurements, denoted $\boldsymbol{d}$, are corrupted by additive noise and are related to the basal sliding coefficient through

$$\boldsymbol{d} = \mathcal{F}(\beta) + \boldsymbol{e}, \tag{6}$$

where $\mathcal{F} : L^2(\Omega) \rightarrow \mathbb{R}^q$ is called the *parameter-to-observable map*, and $\boldsymbol{e} \in \mathbb{R}^q$ denotes the noise in the measurements.

The measurements consist of (noisy) point-wise observations of the velocity field on the top surface. In discrete settings, we compute $\mathcal{F}(\boldsymbol{\beta})$ by first solving the Stokes equations (1) and then applying a linear observation operator that extracts the velocity at the measurement locations. We assume the noise, $\boldsymbol{e}$, is independent of the basal sliding coefficient, has zero mean, and is Gaussian, i.e., $\boldsymbol{e} \sim \mathcal{N}(\boldsymbol{0}, \boldsymbol{\Gamma}_e)$. The likelihood is then of the form (e.g., Bui-Thanh et al., 2013; Petra et al., 2014; Villa

et al., 2020)Tarantola05, KaipioSomersalo05

$$\pi_{\text{like}}(\boldsymbol{d}|\boldsymbol{\beta}) \propto \exp\left\{ -\frac{1}{2} \|\mathcal{F}(\boldsymbol{\beta}) - \boldsymbol{d}\|^2_{\boldsymbol{\Gamma}_e^{-1}} \right\}. \tag{7}$$

### 3.3 The posterior

By applying Bayes' theorem, the posterior density of $\boldsymbol{\beta}$ is proportional to the product of the prior density (5) and the data likelihood (7). This is given by

$$\pi_{\text{post}}(\boldsymbol{\beta}|\boldsymbol{d}) \propto \exp\left\{ -\frac{1}{2} \|\mathcal{F}(\boldsymbol{\beta}) - \boldsymbol{d}\|^2_{\boldsymbol{\Gamma}_e^{-1}} - \frac{1}{2} \|\boldsymbol{\beta} - \boldsymbol{\beta}_*\|^2_{\boldsymbol{\Gamma}_{\text{pr}}^{-1}} \right\}. \tag{8}$$

The corresponding MAP estimate is then defined as

$$\boldsymbol{\beta}_{\text{MAP}} := \operatorname*{argmin}_{\boldsymbol{\beta} \in \mathbb{R}^m} \frac{1}{2} \|\mathcal{F}(\boldsymbol{\beta}) - \boldsymbol{d}\|^2_{\boldsymbol{\Gamma}_e^{-1}} + \frac{1}{2} \|\boldsymbol{\beta} - \boldsymbol{\beta}_*\|^2_{\boldsymbol{\Gamma}_{\text{pr}}^{-1}}. \tag{9}$$

We note that the problem of finding the MAP estimate, defined in (9), reduces to a deterministic inverse problem. To solve this problem we use an inexact Newton conjugate gradient (CG) method, as in Petra et al. (2012). To derive the required first

(i.e., gradient) and second (i.e., Hessian) derivative information needed by Newton's method, we use an adjoint-based method, and refer the reader to Petra et al. (2012) for the derivation and expressions of the required derivatives.

### 3.4 Quantifying posterior uncertainty

To (approximately) quantify the resulting uncertainty in the inferred basal sliding parameter, we invoke a local Gaussian approximation of the posterior (i.e., the Laplace approximation). That is, the solution to the Bayesian inverse problem is

now given by a Gaussian distribution with mean $\boldsymbol{\beta}_{\text{MAP}}$ and covariance $\boldsymbol{\Gamma}_{\text{po}}$ given by the inverse of the (Gauss-Newton) Hessian of the negative log-posterior, evaluated at the MAP estimate. More specifically, we make the approximation, $\boldsymbol{\beta}|\boldsymbol{d} \sim \mathcal{N}(\boldsymbol{\beta}_{\text{MAP}}, \boldsymbol{\Gamma}_{\text{po}})$, with $\boldsymbol{\beta}_{\text{MAP}}$ given by (9), and

$$\boldsymbol{\Gamma}_{\text{po}} = \boldsymbol{H}(\boldsymbol{\beta}_{\text{MAP}})^{-1} = (\overline{\boldsymbol{H}}(\boldsymbol{\beta}_{\text{MAP}}) + \boldsymbol{\Gamma}_{\text{pr}}^{-1})^{-1} = (\boldsymbol{F}^T(\boldsymbol{\beta}_{\text{MAP}})\boldsymbol{\Gamma}_e^{-1}\boldsymbol{F}(\boldsymbol{\beta}_{\text{MAP}}) + \boldsymbol{\Gamma}_{\text{pr}}^{-1})^{-1}, \tag{10}$$



where $\overline{\boldsymbol{H}}(\beta)$ is the Gauss-Newton Hessian of the data misfit term (i.e., the negative log-likelihood), and $\boldsymbol{F}$ is the Jacobian matrix of the parameter-to-observable map, $\mathcal{F}$ (e.g., Bui-Thanh et al., 2013).

The construction of the posterior covariance matrix (i.e., the inverse of the Hessian) is prohibitive for large-scale problems since its dimension is equal to the dimension of the parameter. To make operations with the posterior covariance matrix tractable, we exploit the fact that the eigenvalues of $\overline{\boldsymbol{H}}(\boldsymbol{\beta}_{\mathrm{MAP}})$ collapse to zero rapidly, since the data contain limited information about the (infinite-dimensional) parameter field. Thus a low-rank approximation of the data misfit component of the Hessian $\overline{\boldsymbol{H}}$ can be constructed as in Flath et al. (2011); Bui-Thanh et al. (2013); Petra et al. (2014) by solving the generalized eigenvalue problem

$$\overline{\boldsymbol{H}}\boldsymbol{V}_r = \boldsymbol{\Gamma}_{\mathrm{pr}}^{-1}\boldsymbol{\Lambda}_r\boldsymbol{V}_r, \tag{11}$$

where $\boldsymbol{\Lambda}_r = \mathrm{diag}(\lambda_1, \lambda_2, \ldots, \lambda_r) \in \mathbb{R}^{r \times r}$ is a diagonal matrix collecting the $r$ largest generalized eigenvalues, $\lambda_i$, and $\boldsymbol{V}_r = [\boldsymbol{v}_1, \boldsymbol{v}_2, \ldots, \boldsymbol{v}_r] \in \mathbb{R}^{m \times r}$ is the matrix collecting the corresponding $\boldsymbol{\Gamma}_{\mathrm{pr}}^{-1}$-orthonormal eigenvectors, $\boldsymbol{v}_i$. Above, the truncation index $r$ is chosen such that the remaining eigenvalues, $\lambda_i$, for $i = r+1, \ldots, m$, are *sufficiently* smaller than one.

Substituting $\overline{\boldsymbol{H}} \approx \boldsymbol{V}_r \boldsymbol{\Lambda}_r \boldsymbol{V}_r^T$ into (10) and using the Sherman-Morrison-Woodbury identity (Golub and Van Loan, 1996), after a few algebraic manipulations (e.g., Isaac et al., 2015a), we obtain the following low rank-based approximation of the posterior covariance (under the Laplace approximation)

$$\boldsymbol{\Gamma}_{\mathrm{po}} \approx \boldsymbol{\Gamma}_{\mathrm{pr}} - \boldsymbol{V}_r \boldsymbol{D}_r \boldsymbol{V}_r^T, \tag{12}$$

where $\boldsymbol{D}_r = \mathrm{diag}(\lambda_1/(\lambda_1 + 1), \lambda_2/(\lambda_2 + 1), \ldots, \lambda_r/(\lambda_r + 1)) \in \mathbb{R}^{r \times r}$.

## 4 Premarginalization over auxiliary parameters and the Bayesian approximation error approach

The Bayesian approximation error (BAE) approach (Kaipio and Somersalo, 2007, 2005; Kaipio and Kolehmainen, 2013) can be used to approximately premarginalize over auxiliary parameters. The BAE approach essentially combines all uncertainties, including those generated by fixing uncertain parameters, into a single additive *total error* term. The total error term can then be premarginalized over, i.e., marginalized over before the acquisition of data. We next outline the process.

We denote by $a$ the auxiliary parameters, which in the current study are defined over the entire computational domain, $\Omega$, and are assumed to be Gaussian distributed with covariance operator $\mathcal{C}_a = \mathcal{L}^{-2}$, where $\mathcal{L}$ is defined by

$$\mathcal{L}a := -\nabla \cdot (\gamma_a \nabla a) + \delta_a a, \quad \text{in } \Omega, \tag{13}$$

and mean value $a_*$. Inline with the forward problem, $\mathcal{L}$ is equipped with periodic boundary conditions on the lateral boundaries of $\Omega$, while on the top and bottom boundaries we enforce Robin boundary conditions. Note that explicit knowledge of the distribution of $a$ is not needed, we only require the ability to sample realizations of $a$. In what follows we denote by $\boldsymbol{a}$ any (possibly more than one) discretized auxiliary (uncertain) parameter, such as the rheology parameters.

Next, we let

$$(\boldsymbol{\beta}, \boldsymbol{a}) \mapsto \tilde{\mathcal{F}}(\boldsymbol{\beta}, \boldsymbol{a}) \tag{14}$$





denote an accurate parameter-to-observable mapping, so that the relationship between the parameters and the measured data is given by

$$\boldsymbol{d} = \tilde{\mathcal{F}}(\boldsymbol{\beta}, \boldsymbol{a}) + \boldsymbol{e}. \tag{15}$$

Then, with the aim of avoiding so-called *joint inversion*, i.e., estimating $\boldsymbol{\beta}$ and $\boldsymbol{a}$ simultaneously, we introduce the approxima-
tive parameter-to-observable mapping,

$$\boldsymbol{\beta} \mapsto \mathcal{F}(\boldsymbol{\beta}) = \tilde{\mathcal{F}}(\boldsymbol{\beta}, \boldsymbol{a}_*). \tag{16}$$

That is, the approximative parameter-to-observable map, is the accurate parameter-to-observable map, but with the auxiliary parameters fixed to the associated mean value, i.e., $\boldsymbol{a} = \boldsymbol{a}_*$. Fixing $\boldsymbol{a}$ to some other nominal value is also possible.

The goal is then to carry out estimation of $\boldsymbol{\beta}$ using only the approximative parameter-to-observable map, $\mathcal{F}(\boldsymbol{\beta})$, while taking
into account the (statistics of) the discrepancy between the models. To this end, equation (15) is reformulated as

$$\boldsymbol{d} = \tilde{\mathcal{F}}(\boldsymbol{\beta}, \boldsymbol{a}) + \boldsymbol{e} = \mathcal{F}(\boldsymbol{\beta}) + \boldsymbol{e} + \boldsymbol{\varepsilon} = \mathcal{F}(\boldsymbol{\beta}) + \boldsymbol{\nu}, \tag{17}$$

where $\boldsymbol{\varepsilon} = \tilde{\mathcal{F}}(\boldsymbol{\beta}, \boldsymbol{a}) - \mathcal{F}(\boldsymbol{\beta})$ is known as the *approximation error* and $\boldsymbol{\nu}$ as the total error (e.g., Nicholson et al., 2018; Tarvainen et al., 2010). Next, the approximation error is approximated as Gaussian, $\boldsymbol{\varepsilon} \sim \mathcal{N}(\boldsymbol{\varepsilon}_*, \boldsymbol{\Gamma}_\varepsilon)$. Though, formally, the approximation error depends on the parameters, i.e., $\boldsymbol{\varepsilon} = \boldsymbol{\varepsilon}(\boldsymbol{\beta}, \boldsymbol{a})$, a further approximation, termed the *enhanced error model* or the *composite*
*error model* approximation, is often made, which approximates $\boldsymbol{\varepsilon}$ as independent of all parameters (Kaipio and Kolehmainen, 2013). This leads to the total errors being distributed as $\boldsymbol{\nu} \sim \mathcal{N}(\boldsymbol{\nu}_*, \boldsymbol{\Gamma}_\nu) = \mathcal{N}(\boldsymbol{\varepsilon}_*, \boldsymbol{\Gamma}_e + \boldsymbol{\Gamma}_\varepsilon)$.

Use of the BAE approach results in an updated posterior density for $\boldsymbol{\beta}$;

$$\pi_{\mathrm{post}}^{\mathrm{BAE}}(\boldsymbol{\beta}) \propto \exp\left\{ -\frac{1}{2} \|\mathcal{F}(\boldsymbol{\beta}) - \boldsymbol{d} + \boldsymbol{\nu}_*\|_{\boldsymbol{\Gamma}_\nu^{-1}}^2 - \frac{1}{2} \|\boldsymbol{\beta} - \boldsymbol{\beta}_*\|_{\boldsymbol{\Gamma}_{\mathrm{pr}}^{-1}}^2 \right\}, \tag{18}$$

which is obtained by explicit marginalization over $\boldsymbol{\nu}$ (Kaipio and Kolehmainen, 2013). The updated MAP estimate is then

$$\boldsymbol{\beta}_{\mathrm{MAP}}^{\mathrm{BAE}} := \underset{\boldsymbol{\beta} \in \mathbb{R}^m}{\operatorname{argmin}} \left\{ \frac{1}{2} \|\mathcal{F}(\boldsymbol{\beta}) - \boldsymbol{d} + \boldsymbol{\nu}_*\|_{\boldsymbol{\Gamma}_\nu^{-1}}^2 + \frac{1}{2} \|\boldsymbol{\beta} - \boldsymbol{\beta}_*\|_{\boldsymbol{\Gamma}_{\mathrm{pr}}^{-1}}^2 \right\}. \tag{19}$$

This updated expression for the MAP estimate is only a slight modification of the original MAP estimate given in (9), thus reformulating the corresponding adjoint, incremental forward, and incremental adjoint equations is essentially trivial. Lastly, the updated posterior covariance matrix (under the Laplace approximation) is now given by

$$\boldsymbol{\Gamma}_{\mathrm{po}}^{\mathrm{BAE}} = (\boldsymbol{F}^T(\boldsymbol{\beta}_{\mathrm{MAP}}) \boldsymbol{\Gamma}_\nu^{-1} \boldsymbol{F}(\boldsymbol{\beta}_{\mathrm{MAP}}) + \boldsymbol{\Gamma}_{\mathrm{pr}}^{-1})^{-1}. \tag{20}$$

## 4.1    Computing the approximation error statistics

In the current paper, all parameters are taken to have Gaussian (prior) distributions, i.e., $z \sim \mathcal{N}(z_*, \mathcal{C}_z)$, with $z = (\beta, a)$. We also assume $\beta$ and $a$ are independent, thus specifying $\beta_*, a_*, \mathcal{C}_\beta$, and $\mathcal{C}_a$ fully describes the prior density.





Unlike the statistics of the parameters and the measurement noise, the mean and covariance of the approximation errors, $\varepsilon_*$ and $\mathbf{\Gamma}_\varepsilon$ respectively, must in general be estimated based on (Monte Carlo) samples. That is,

$$\varepsilon_* = \frac{1}{N} \sum_{\ell=1}^{N} \varepsilon^{(\ell)}, \quad \text{and} \quad \mathbf{\Gamma}_\varepsilon = \frac{1}{N-1} \boldsymbol{E} \boldsymbol{E}^T, \tag{21}$$

with $N \in \mathbb{N}$ the number of samples, $\varepsilon^{(\ell)} = \tilde{\mathcal{F}}(\boldsymbol{\beta}^{(\ell)}, \boldsymbol{a}^{(\ell)}) - \mathcal{F}(\boldsymbol{\beta}^{(\ell)})$, for $\ell = 1, 2, \dots, N$, where $\boldsymbol{\beta}^{(\ell)}$ and $\boldsymbol{a}^{(\ell)}$ are samples drawn from the joint prior density, and $\boldsymbol{E} = [\varepsilon^{(1)} - \varepsilon_*, \varepsilon^{(2)} - \varepsilon_*, \dots, \varepsilon^{(N)} - \varepsilon_*]$.

It's worth noting, that all sampling and computations of the approximation errors and the associated statistics can be carried out prior to the acquisition of any data, and is thus often termed *offline computations* (Kaipio and Kolehmainen, 2013). Furthermore, though the computational cost per sample of $\varepsilon$ in the current paper is two forward (nonlinear) Stokes solves, the sampling procedure is *embarrassingly parallel*, i.e., each sample can be carried out independently, and in practice, only a fairly small number of samples is required.

We conclude this section by giving several *rules of thumb* relating to the use of the BAE approach, for more details see Kaipio and Kolehmainen (2013). Firstly, the total number of samples required to accurately construct the statistics of the approximation errors is generally (often substantially) less than $N = 1000$. Secondly, if

$$\text{trace}(\mathbf{\Gamma}_e) < \|\varepsilon_*\|^2 + \text{trace}(\mathbf{\Gamma}_\varepsilon) \tag{22}$$

holds, then the approximation errors are said to *dominate* the noise, and neglecting them will generally result in misleading, and potentially infeasible, results. Finally, if for any $\boldsymbol{w} \in \mathbb{R}^q$,

$$\boldsymbol{w}^T \mathbf{\Gamma}_e \boldsymbol{w} < (\boldsymbol{w}^T \varepsilon_*)^2 + \boldsymbol{w}^T \mathbf{\Gamma}_\varepsilon \boldsymbol{w}, \tag{23}$$

then the induced modeling errors should also be accounted for.

## 5 Numerical examples

In this section, we outline three numerical examples to assess the applicability, performance, and robustness of the BAE approach to account for uncertain rheology parameters. In all cases the parameter of interest is the basal sliding coefficient, $\beta$. Any other unknown/uncertain parameters are (approximately) premarginalised over using the BAE approach, as outlined in Section 4.

The forward problems considered here are based on the models used in the Ice Sheet Model Intercomparison Project for Higher-Order Models (ISMIP-HOM) benchmark study carried out in Pattyn et al. (2008). Accordingly, all problems are considered in box-like geometries, i.e., $\Omega = [0, L]^{d-1} \times [0, H]$, for $d = 2$ (in Examples 1 and 2) or $d = 3$ (in Example 3). Furthermore, in all model problems we take the ice slab to be set on an incline plane with slope $\theta = 0.1°$, the density of the ice to be $\rho = 910 \text{ kg m}^{-d}$, and the gravitational acceleration constant to be $g = 9.81 \text{ m s}^{-2}$. For all examples we set the length at $L = 10$km, while for Examples 1 and 2, we set $H = 250$ m, and for Example 3, we set $H = 1$ km. In Fig. 1 we show a two-dimensional schematic of the model problems set up.





The true basal sliding coefficient fields used for each example are based on those in Petra et al. (2012). Specifically, letting $\omega = 2\pi/L$, for Examples 1 and 2 (posed in two dimensions) we set

$$\beta(s) = 7 + \sin(\omega s), \quad \forall s \in \Gamma_b, \tag{24}$$

as shown in Fig. 7, while in Example 3 (posed in three dimensions) we set

$$\beta(\boldsymbol{s}) = 7 + 3\sin(\omega x)\sin(\omega y), \quad \forall \boldsymbol{s} = (x, y) \in \Gamma_b, \tag{25}$$

as shown in Fig. 13.

For all numerical experiments we use synthetic measurements; these are randomly placed noisy point-wise measurements of each component of the velocity on the top surface of the domain, i.e., at points on $\Gamma_t$. Examples 1 and 2 are carried out based on $q = 80$ measurement locations, while for Example 3 we use $q = 100$ measurement locations. These measurements are obtained by adding zero mean white noise to the solution of the forward problem. Thus the additive noise is of the form $\boldsymbol{e} \sim \mathcal{N}(\boldsymbol{0}, \boldsymbol{\Gamma}_e)$ with covariance matrix $\boldsymbol{\Gamma}_e = \delta_e^2 \boldsymbol{I}$. We take $\delta_e$ to satisfy $\delta_e = (1/100) \times (\max(\mathcal{B}\boldsymbol{u}(\beta_{\text{true}})) - \min(\mathcal{B}\boldsymbol{u}(\beta_{\text{true}})))$, i.e., the noise level is 1% of the range of the noiseless synthetic measurements. The precise noise level is problem specific, however, when using GPS techniques and InSAR velocity measurements, a 1% noise level is realistic; see for example Martin and Monnier (2014) and the references therein.

For all examples considered here, the prior mean for the basal sliding coefficient, $\beta$, is set at $\beta_* = 7$. On the other hand, the prior covariance operator, $\mathcal{C}_\beta$, is identical for Examples 1 and 2, while for Example 3 different controlling parameters are used, details are provided in Table 1. Along with the true basal sliding coefficient used in Examples 1 and 2, we also show the prior distribution and three samples drawn from the prior in Fig. 7. In Fig. 12, we show four samples from the prior used for Example 3.

## 5.1 Example problems

We now give the specific details of each model problem, and make apparent which parameters we treat as auxiliary parameters, and subsequently premarginalize over. Key details about each model problem are summarized in Table 1.

**Example 1: Uncertain flow rate factor in the two-dimensional linear Stokes ice sheet model**

The first example is carried out assuming a linearized (Stokes) ice sheet model in two-dimensions. Specifically, we set $n = 1$ in (1), resulting in the effective viscosity being given by $\eta(\boldsymbol{x}) = \frac{1}{2}A(\boldsymbol{x})^{-1}$. The flow rate factor, $A$, is taken to be unknown and spatially varying, as is often the case in reality. We represent the flow rate factor as $A = A_0 \exp(-a(\boldsymbol{x}))$, with $A_0 = 2.140373 \times 10^{-7} \, \text{Pa}^{-1} \, \text{a}^{-1}$, and the *pre-factor*, $\exp(-a(\boldsymbol{x}))$, taking the role of the auxiliary parameter, which will subsequently be premarginalized over using the BAE approach. The pre-factor accounts for several physical and computational phenomena, such as the Arrhenius relationship between $A(\boldsymbol{x})$ and the ice temperature (e.g., Cuffey and Paterson, 2010; Zhu et al., 2016), and the use of so-called *enhancement factors* (Cuffey and Paterson, 2010; Ma et al., 2010). The 'exp' function is used to ensure the pre-factor remains positive.





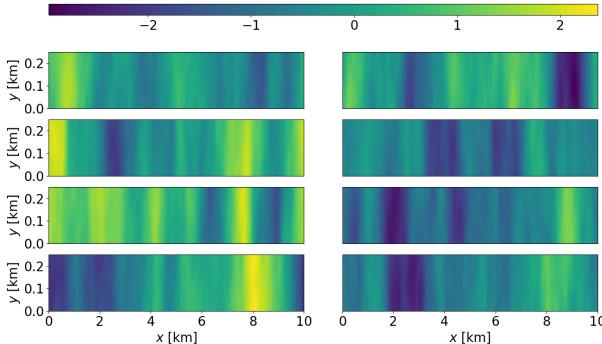

**Figure 2.** Samples of the flow rate pre-factor for Example 1. Left column: Samples for Example 1a. Right column: Samples for Example 1b. The samples in the top row are taken as the true flow rate pre-factors. Note that the axes have been stretched in the $y$-direction for ease of visualization.

The prior distribution of the flow rate pre-factor is set by taking the prior mean to be $a_* = 0$, while the controlling parameters of the prior covariance operator are given in Table 1. The true pre-factor and three draws from the associated prior distribution are shown in Fig. 2 for Examples 1a and 1b. As outlined below, the computational meshes used for Examples 1a and 1b are different. This leads to differences in the true pre-factor used for both examples. This in turn results in different synthetic data being used for the inversions, however, in both cases the standard deviation of the noise is $\delta_e \approx 0.07$. In both cases the flow rate pre-factor is discretized using continuous quadratic Lagrange basis functions.

We use this example to also demonstrate that the proposed approach is independent of the discretization, a critical property to have when aiming to solve large-scale problems. This is done by considering identical problems on two different levels of discretization. Specifically, we consider the problem on two structured meshes having substantially different levels of discretization:

a) In the first case, the computational mesh consists of 2000 triangular elements, which results in the discretized velocity and pressure having 8400 degrees of freedom (dofs), and 1100 dofs while the basal sliding coefficient has 100 unknowns, and the flow rate pre-factor has 4200 dofs.

b) In the second case, the mesh is refined and it consists of 8000 triangular elements, leading to 32800, and 4200 dofs for the discretized velocity and pressure, respectively, while the dimensions of the basal sliding coefficient and the flow rate pre-factor are 200 and 16400, respectively.

**Example 2: Uncertain Glen's flow law exponent in the two-dimensional nonlinear Stokes ice sheet model**

For the second example we use the nonlinear Stokes problem (1) as the governing equation. We take the Glen's flow law exponent, $n(\boldsymbol{x})$, as an uncertain (and unknown) spatially varying auxiliary parameter, i.e., we set $a(\boldsymbol{x}) = n(\boldsymbol{x})$, and proceed to approximately premarginalize over it. The prior mean of the Glen's flow law exponent is set to $a_* = 3$, while the parameters controlling the covariance operator, $\mathcal{C}_a$, are given in Table 1, and are chosen to ensure that the Glen's flow law exponents are



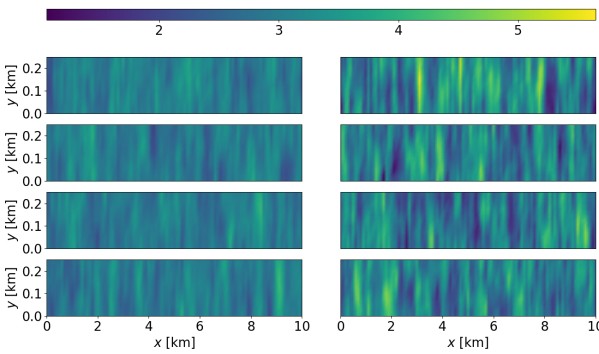

**Figure 3.** Samples of the Glen's flow law exponent for Example 2. Left column: Samples for Example 2a. Right column: Samples for Example 2b. The samples in the top row are taken as the true Glen's flow law exponents. Note that the axes have been stretched in the $y$-direction for ease of visualization.

inline with the literature. Furthermore, to ensure shear-thinning we enforce $1 \leq n(\boldsymbol{x})$. This is done by *rejection sampling*, and corresponds to constraining the function space in which $n$ lies, see (Dashti and Stuart, 2016, Equation (10.10)) for details.

We also use this example to study the effect of larger modeling errors (i.e., *excessive* errors). That is, we consider the case when the variance of the approximation errors is so large that essentially all information in the data is *washed out*. As we shall

see, however, the resulting uncertainty estimates are still feasible. To induce larger uncertainties (and resulting approximation errors) we alter the prior covariance operator for the Glen's flow law exponent, $n$, to favor more highly oscillatory realizations. We can thus further divide Example 2 into two cases:

a) The case of modest approximation errors.

b) The case of excessive approximation errors.

The parameters used to control the covariance of the distributions on $n$ are shown in Table 1. The true Glen's flow law exponents used to generate the data for Examples 2a and 2b are drawn from the respective distributions, which, along with several other samples of the Glen's flow law exponent from each of the distributions, are shown in Fig. 3. In both cases the Glen's flow law exponent is discretized using continuous linear Lagrange basis function, while the computational mesh used is the same as that used in Example 1a. Finally, in Example 2a we have $\delta_e \approx 0.04$ while in Example 2b we have $\delta_e \approx 0.05$.

**Example 3: Uncertain flow rate factor in the three-dimensional nonlinear Stokes ice sheet model**

In this example, we consider a three-dimensional ($d = 3$), nonlinear analogue of Example 1. Specifically, we consider (1) in three dimensions, with the Glen's flow law exponent set to $n = 3$. Similarly to Example 1, we suppose the flow rate factor is spatially heterogeneous, unknown, and parameterized as $A = A_0 \exp(-3a(\boldsymbol{x}))$. The nominal value for the flow rate factor is set at $A_0 = 10^{-16}$ Pa$^{-3}$ a$^{-1}$, while the pre-factor, $\exp(-3a(\boldsymbol{x}))$, takes into account several physical and computational

phenomena as described previously.



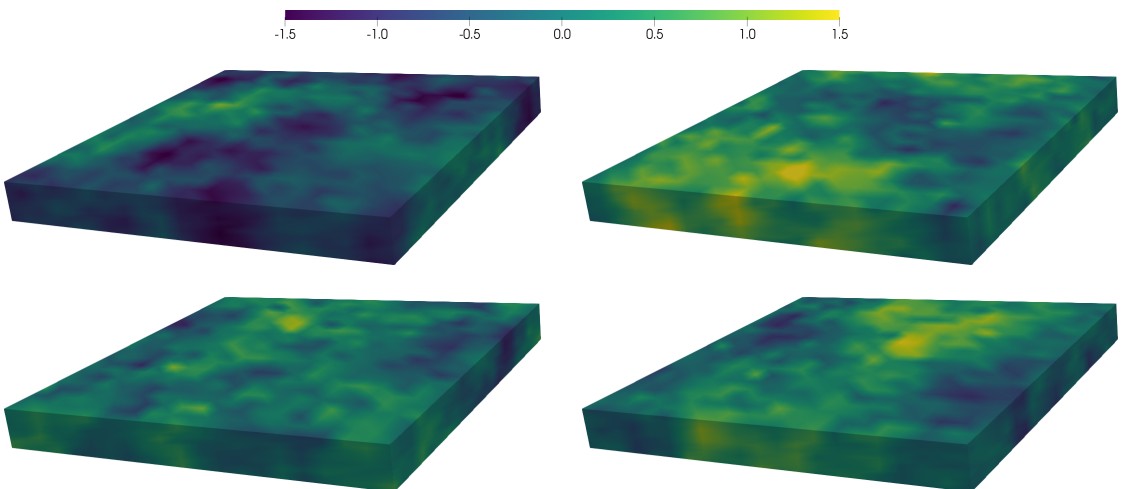

**Figure 4.** Samples of the flow rate pre-factor for Example 3. The top-left sample is taken as the true flow rate pre-factor. Note that the domain has been stretched in the $z$-direction for ease of visualization.

The mean value of the auxiliary parameter is set at $a_* = 0$, while the parameters controlling the distribution of the pre-factor are given in Table 1. These values for the prior covariance operator of $a$ ensure the flow rate values are inline with those presented in the literature, see for example Table 3.4 of Cuffey and Paterson (2010). In Fig. 4, we show the true flow rate pre-factor along with three samples from the associated prior density. Unlike Example 1, the flow rate pre-factor in this example

is discretized using continuous linear Lagrange basis functions. The computational mesh used consists of 19200 tetrahedral elements, leading to 81600 dofs for the velocity, 3600 dofs for the pressure, 27200 for the flow rate pre-factor, and 400 for the basal sliding coefficient.

## 5.2   Estimates and approximate posterior covariances

For each of the examples listed above, we compare the estimation results (MAP points and approximate posterior covariances)

for three different approaches. Within each example, for each of the approaches, the same prior distribution is used for the basal sliding coefficient, thus it is only the associated likelihoods that differ. In our analysis we place particular emphasis on the feasibility of the posterior estimates, that is, whether or not the computed posterior distributions support the true basal sliding coefficient. The three different approaches considered are:

a) **The accurate case (REF):** in this case any auxiliary parameters are set to their true values, i.e., we use $\tilde{\mathcal{F}}(\beta, a_{\mathrm{true}})$ as the

parameter-to-observable map. REF is computed as a benchmark/reference. The resulting likelihood for REF is

$$\pi^{\mathrm{REF}}(\boldsymbol{d}|\boldsymbol{\beta}) \propto \exp\left\{ -\frac{1}{2} \left\| \tilde{\mathcal{F}}(\boldsymbol{\beta}, \boldsymbol{a}_{\mathrm{true}}) - \boldsymbol{d} \right\|^2_{\boldsymbol{\Gamma}_e^{-1}} \right\}, \tag{26}$$

while the accurate MAP estimate and the corresponding posterior covariance matrix are denoted by $\boldsymbol{\beta}^{\mathrm{REF}}_{\mathrm{MAP}}$ and $\boldsymbol{\Gamma}^{\mathrm{REF}}_{\mathrm{po}}$, respectively.





**Table 1.** Details for each of the examples considered. The first column (Ex.) refers to the example number; the second, third, and fourth columns give details of the forward model used, including which Stokes model is used, the aspect ratio, $L/H$, and the definition of the auxiliary parameter; the fifth, sixth, and seventh columns give the discretization details, including the number of degrees of freedom for the velocities and pressure, the number of degrees of freedom of the unknown parameters ($(\beta, a)$ dofs), and the number of measurements, $q$; finally, the eighth through twelfth columns give details on the prior distributions for the unknowns, including the parameters controlling the prior covariance operator for $\beta$, the controlling parameters for the prior covariance operator of $a$, and the prior mean, $a_*$. Note that in Examples 2a and 2b the prior for the auxiliary parameter is further constrained by enforcing $1 \leq n(\boldsymbol{x})$, while for all examples the prior mean for the basal sliding coefficient is taken as $\beta_* = 7$.

| | Model details | | | Discretization details | | | Prior distribtion details | | | | |
|---|---|---|---|---|---|---|---|---|---|---|---|
| Ex. | Stokes | $L/H$ | $a$ | $(\boldsymbol{u}, p)$ dofs | $(\beta, a)$ dofs | $q$ | $\gamma_\beta$ | $\delta_\beta$ | $a_*$ | $\gamma_a$ | $\delta_a$ |
| 1a | Linear 2D | 40 | $\ln(A_0/A)$ | (8400, 1100) | (100, 4200) | 80 | 840 | $7.0 \times 10^{-5}$ | 0 | 300 | $1.5 \times 10^{-4}$ |
| 1b | Linear 2D | 40 | $\ln(A_0/A)$ | (32800, 4200) | (200, 16400) | 80 | 840 | $7.0 \times 10^{-5}$ | 0 | 300 | $1.5 \times 10^{-4}$ |
| 2a | Nonlinear 2D | 40 | $n$ | (8400, 1100) | (100, 1100) | 80 | 840 | $7.0 \times 10^{-5}$ | 3 | 90 | $9.0 \times 10^{-3}$ |
| 2b | Nonlinear 2D | 40 | $n$ | (8400, 1100) | (100, 1100) | 80 | 840 | $7.0 \times 10^{-5}$ | 3 | 41 | $4.1 \times 10^{-3}$ |
| 3 | Nonlinear 3D | 10 | $\frac{1}{3}\ln(A_0/A)$ | (81600, 3600) | (400, 27200) | 100 | 7.5 | $7.5 \times 10^{-7}$ | 0 | 12.5 | $2.5 \times 10^{-6}$ |

b) **The conventional error model approach (CEM):** this approach uses the standard error model (induced by the additive error, $\boldsymbol{e}$), while using the approximative model, $\mathcal{F}(\boldsymbol{\beta})$, where the auxiliary parameters are set to some nominal value (such as $a = a_*$). The likelihood is then of the form

$$\pi^{\text{CEM}}(\boldsymbol{d}|\boldsymbol{\beta}) \propto \exp\left\{-\frac{1}{2}\|\mathcal{F}(\boldsymbol{\beta}) - \boldsymbol{d}\|^2_{\boldsymbol{\Gamma}_e^{-1}}\right\}. \tag{27}$$

5    We denote the corresponding MAP estimate and the posterior covariance matrix by $\boldsymbol{\beta}^{\text{CEM}}_{\text{MAP}}$ and $\boldsymbol{\Gamma}^{\text{CEM}}_{\text{po}}$, respectively.

c) **The Bayesian approximation error approach (BAE):** this approach also uses the approximative model, $\mathcal{F}(\boldsymbol{\beta})$, but accounts for the approximation errors using the BAE approach outlined in Section 4. As given in (19), the updated likelihood found using the BAE approach is

$$\pi^{\text{BAE}}(\boldsymbol{d}|\boldsymbol{\beta}) \propto \exp\left\{-\frac{1}{2}\|\mathcal{F}(\boldsymbol{\beta}) - \boldsymbol{d} + \boldsymbol{\nu}_*\|^2_{\boldsymbol{\Gamma}_\nu^{-1}}\right\}, \tag{28}$$

10    with the MAP estimate and the posterior covariance matrix denoted by $\boldsymbol{\beta}^{\text{BAE}}_{\text{MAP}}$ and $\boldsymbol{\Gamma}^{\text{BAE}}_{\text{po}}$, respectively.

# 6   Results

Here we discuss and compare MAP estimates for the basal sliding coefficient and the respective approximate posterior covariance for each example. As alluded to previously, we pay particular attention to the feasibility of the posterior uncertainty estimates when comparing the results. We also examine the spectrum of the prior preconditioned misfit Hessians, which gives





further insight into the uncertainty, and the sensitivity of each approach. To conclude the section, we give a brief comparison of the online computational costs (in terms of linearized Stokes PDE solves) for computing the MAP estimates.

To solve the optimization problems we use an inexact Newton-CG method, see for example Petra et al. (2012). In all cases we start the optimization procedure using the prior mean for the initial estimate of the basal sliding coefficient, while the prior

covariance operator is used as the preconditioner. The optimization is carried out using Gauss-Newton Hessian approximation for the first five iterations and then full Newton, combined with an Armijo linesearch (Nocedal and Wright, 2006). Convergence is established when the gradient has decreased by a factor of $10^6$, relative to the norm of the initial gradient.

The numerical results presented in this paper are obtained using hIPPYlib (an inverse problem Python library (Villa et al., 2018; Villa et al., 2020)). hIPPYlib implements state-of-the-art scalable adjoint-based algorithms for PDE-based deterministic

and Bayesian inverse problems. It builds on FEniCS (Dupont et al., 2003; Logg et al., 2012) for the discretization of the PDEs and on PETSc (Balay et al., 2001, 2009) for scalable and efficient linear algebra operations and solvers needed for the solution of the PDEs. Inline with the finite element discretization used for the weak form of the forward problem (2), in what follows we use Taylor-Hood finite elements for the adjoint, incremental forward, and incremental adjoint equations, as in Petra et al. (2012).

## 6.1 Example 1

In this example, we consider the case of an uncertain flow rate factor in the two-dimensional linear Stokes ice sheet model, and demonstrate the mesh independence of the approach. We begin by discussing the statistics of the approximation errors, which are induced by treating the unknown flow rate factor as a known constant, specifically, $A = 2.140373 \times 10^{-7}\mathrm{Pa}^{-1}\mathrm{a}^{-1}$. In Fig. 5, we show the marginal distribution of the approximation errors in the $x$-component (top) and $y$-component (bottom)

for Example 1a (left) and Example 1b (right). The approximation errors are similar for the coarser mesh (Example 1a) and the finer mesh (Example 1b), both having fairly constant mean and variance in each component. For both examples, the mean of the approximation errors in the $x-$component of the velocity measurements is non-zero, $\varepsilon_* \approx 0.2$, while the standard deviation of the approximation errors is substantially larger than the additive noise ($\delta_e \approx 0.07$). That is, the approximation errors dominate the additive noise, as explained in Section 4.1, and it is likely (and is indeed the fact), that ignoring the approximation errors

may lead to infeasible results.

To illustrate the convergence of the approximation errors, in the top row of Fig. 6 we show the spectrum of the approximation errors covariance matrices, $\mathbf{\Gamma}_\varepsilon$, for Examples 1a and 1b for increasing sample sizes ($N = 62, 250, 500$, and 1000). From the figure, it is evident that for $N \geq 250$ samples, the spectra essentially coincide, and have both converged, thus demonstrating the discretization independence of the approach, and that approximately $N \geq 250$ samples is likely sufficient to characterize

the approximation error statistics. Note, however, that the results displayed here use $N = 1000$ samples.

To give further insight into the distribution of the approximation errors, we show the (Pearson's) correlation matrix of the approximation errors for Example 1a (left) and 1b (right) in the bottom row of Fig. 6. Firstly, we notice that the correlation matrices are highly structured (unlike the noise covariance matrix which is diagonal). It is also apparent that the correlation matrices are (visually) identical, further illustrating the discretization independence. The $2 \times 2$ block structure of the correla-


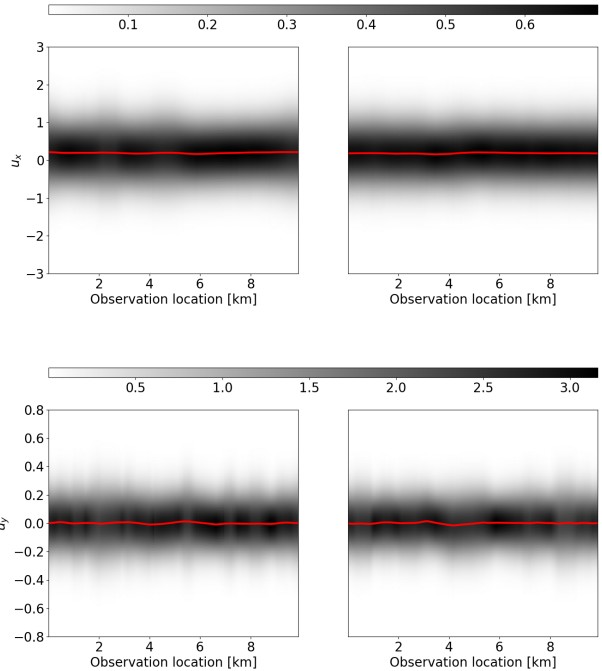

**Figure 5.** Second order statistics of the approximation errors for Example 1. Top row: Distribution of the approximation errors in the $x-$direction velocity measurements for Example 1a (left) and 1b (right). Bottom row: Distribution of the approximation errors in the $y-$direction velocity measurements for Example 1a (left) and 1b (right). The mean of the approximation errors, $\varepsilon_*$, is indicated with a red line, while higher probability density is indicated by darker shading.

tion matrices is to be expected since the measurement number indexing used corresponds to measuring the $q = 80$ velocity measurements in the $x-$direction first, followed by the 80 velocity measurements in the $y-$direction. The behavior within the diagonal blocks is also fairly intuitive as periodic boundary conditions are used, while the structure also illustrates that measurements (relatively) far away from each other are fairly uncorrelated. Comparing the diagonal blocks we see that the approximation errors in the $x-$component of the velocity measurements are more highly correlated at greater distances than those of the $y-$component. Finally, we see that the off-diagonal blocks also display fairly complex behavior.

In the top row of Fig. 7, we show the marginal prior distributions and the resulting marginal posterior distributions. Also shown are the corresponding MAP estimates, the true basal sliding coefficient, and three draws from each of the distributions. Firstly, the accurate MAP estimate, $\beta_{\mathrm{MAP}}^{\mathrm{REF}}$, is in good agreement with the true basal sliding coefficient, while the accurate posterior is clearly feasible in the sense that the true basal sliding coefficient is well supported by the Laplace approximated posterior. On the other hand, the MAP estimate found using the conventional error model, $\beta_{\mathrm{MAP}}^{\mathrm{CEM}}$, differs substantially from the true basal sliding coefficient, over most of the domain. Furthermore, the posterior is essentially infeasible, with the actual coefficient having virtually no posterior density. Conversely, the MAP estimate found using the BAE approach, $\beta_{\mathrm{MAP}}^{\mathrm{BAE}}$, is in fairly good agreement with the true coefficient, and the Laplace approximated posterior supports the truth well. We do see

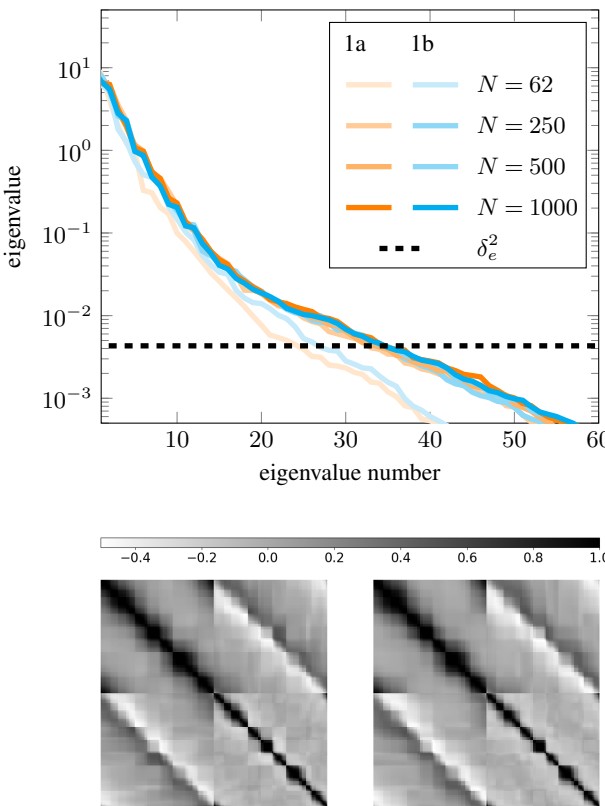

**Figure 6.** Convergence and (Pearson's) correlation matrix of the approximation errors for Example 1. Top row: Spectrum of $\boldsymbol{\Gamma}_\varepsilon$ for various sample sizes, $N$, for Example 1a (orange) and Example 1b (cyan), along with the noise variance, $\delta_e^2$. Bottom row: (Pearson's) correlation matrix of the approximation errors for Example 1a (left) and 1b (right).

that the marginal posterior standard deviations found using the BAE approach are somewhat larger than those found using the accurate and conventional error approaches. This is typical, and to be expected, as the additional uncertainty in the flow rate pre-factor manifest itself as extra posterior uncertainty.

In the bottom row of Fig. 7, we show the corresponding results for Example 1b. The results are fairly similar to Example 1a when using the accurate approach and when using the BAE approach[2] despite the substantial difference in the discretizations used. Lastly, the MAP estimate found using the conventional error model has changed drastically from Example 1a, though the posterior is equally as bad.

---

[2]We attribute the differences in the BAE approach to the differences in the true flow rate pre-factor, the noise realization, and the specific samples of $\boldsymbol{\varepsilon}$.

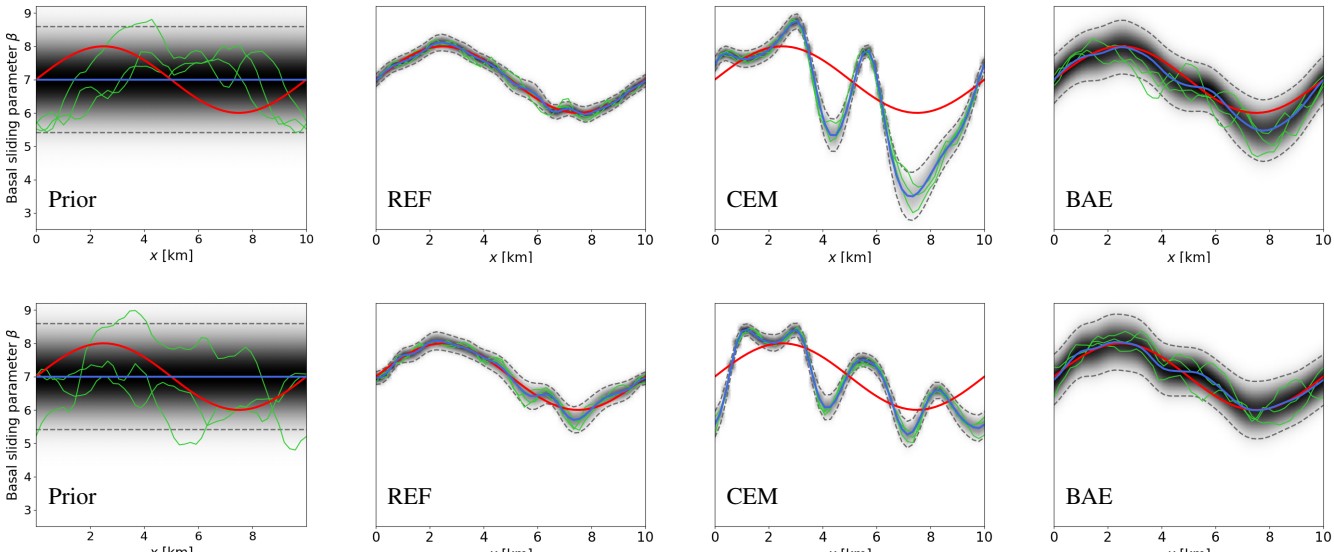

**Figure 7.** Prior and MAP estimates of the basal sliding parameter for Example 1. Top row: Example 1a prior (far left), accurate/reference (REF) case (centre left), conventional error model (CEM) case (centre right), and Bayesian approximation error (BAE) case (far right). Bottom row shows the same plots for Example 1b. In each plot, the mean of the distribution (blue line) is shown along with three samples from the respective distributions (green lines), the marginal distribution (shaded) with darker shading indicating higher probability, and the $\pm 2$ (approximate) standard deviation intervals (black dashed line).

## 6.2 Example 2

In this example, we consider the case of an uncertain Glen's flow law exponent in the two-dimensional nonlinear Stokes ice sheet model, and also demonstrate what happens when the approximation errors are, in some sense, too large. The approximation errors here are the result of treating the unknown, and spatially varying, Glen's flow law exponent as a fixed constant, i.e., setting $n = n_0 = 3$. To induce the larger approximation errors for Example 2b, compared to Example 2a, we increase the uncertainty in the Glen's flow law exponent by altering the associated prior distribution (see Table 1). The difference in magnitude of the approximation errors is apparent in Fig. 8, where we show the marginal distribution of the approximation errors at the observation locations in the $x-$direction (top) and $y-$direction (bottom) velocities for Examples 2a (left) and 2b (right). Note that the variance of the approximation errors for Example 2b is substantially larger than that of Example 2a. Considering that the standard deviation of the added noise for the *small* approximation error case (Example 2a) is $\delta_{e_a} \approx 0.04$ and for the *large* approximation error case (Example 2b) is $\delta_{e_b} \approx 0.05$, the approximation errors in both examples dominate the noise, see Section 4.1.

The top row of Fig. 9 shows the spectrum of the covariance matrices of the approximation errors, $\mathbf{\Gamma}_\varepsilon$, for $N = 125, 500, 1000$, and 2000 samples, for Example 2a, and for $N = 250, 1000, 2000$, and 4000 samples, for Example 2b. For Example 2a, it appears $N \approx 500$ is enough samples, though for the results here we used $N = 1000$, while for Example 2b we require $N \approx 2000$



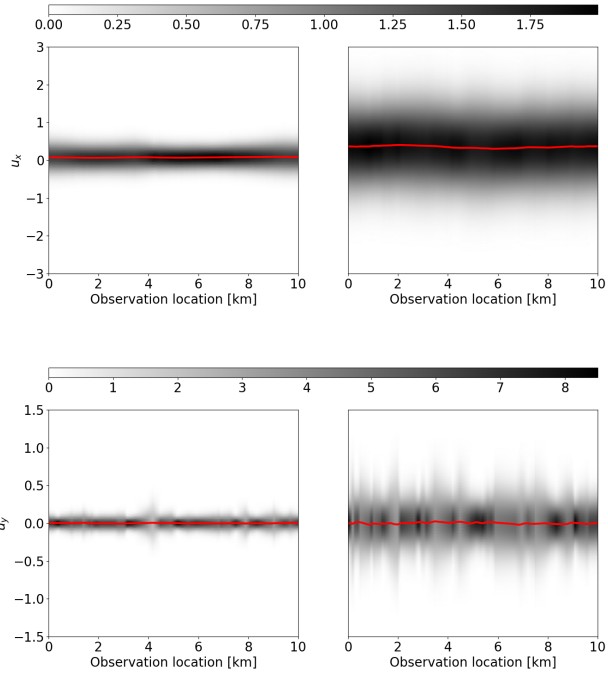

**Figure 8.** Second order statistics of the approximation errors for Example 2. Top row: Distribution of the approximation errors in the $x-$direction velocity measurements for Example 2a (left) and 2b (right). Bottom row: Distribution of the approximation errors in the $y-$direction velocity measurements for Example 2a (left) and 2b (right). The mean of the approximation errors, $\varepsilon_*$, is indicated with a red line, while higher probability density is indicated by darker shading.

samples. The fact that more samples are required to ensure convergence of the approximation errors in Example 2b follows naturally from the increased uncertainty. It's worth pointing out that Example 2b is used mainly to demonstrate how the BAE approach performs in the presence of *too much* modeling uncertainty, thus for the purposes of the current study we deem taking $N = 2000$ as *tolerable*.

5   In the bottom row of Fig. 9, we show the (Pearson's) correlation matrices of the approximation errors. The correlation matrices for this example share several of the characteristics seen in the corresponding correlation matrices in Example 1. Specifically, the block structure, and general behavior within the blocks. Comparing the correlation matrices for Examples 2a and 2b, it appears the approximation errors in the $x-$component for Example 2a are more highly correlated at greater distances towards the edges of the computational domain, compared to the approximation errors in the $x-$component of the velocity

10  measurements for Example 2b.

In the top row of Fig. 10, we show the marginal prior and Laplace approximated posterior distributions, as well as three draws from each of the distributions, the corresponding MAP estimates, and the true basal sliding coefficient for Example 2a. A couple of conclusions can be drawn from this figure. First, the accurate MAP estimate, $\beta_{\mathrm{MAP}}^{\mathrm{REF}}$, closely resembles the true basal sliding coefficient, and the truth is well supported by the accurate posterior distribution. Second, the Laplace approximated posterior



found using the conventional error approach is infeasible for most of the right half of the domain, with the MAP estimate, $\beta_{\text{MAP}}^{\text{CEM}}$, (severely) underestimating the true basal sliding coefficient. Third, the true basal sliding coefficient lies well within the bulk of the (Laplace approximated) posterior for the BAE approach, with the MAP estimate, $\beta_{\text{MAP}}^{\text{BAE}}$, in fairly good agreement with the true basal sliding coefficient.

In the bottom row of Fig. 10, we show the corresponding results for Example 2b, in which the approximation errors are excessive. Under the Laplace approximation, the accurate posterior, found by using the true Glen's flow law exponent, remains an accurate representation of the truth as in Example 2a. The posterior found using the conventional error model approach has significantly deteriorated, however, with the true basal sliding coefficient even more markedly underestimated, and the truth lying well outside the bulk of the Laplace approximated posterior over almost all of the domain. Conversely, by taking into

account the excessive modeling errors in Example 2b, the posterior found using the BAE approach is comparable to the prior, with the corresponding MAP estimate, $\beta_{\text{MAP}}^{\text{BAE}}$, being fairly similar to the prior mean. This demonstrates that when using the BAE approach, as the modeling errors become larger, the corresponding posterior density tends towards the prior, as should be hoped, to avoid overconfidence in biased results.

### 6.3   Example 3

In this example, we consider an uncertain flow rate factor in a larger scale, three-dimensional nonlinear Stokes ice sheet model. The approximation errors are the result of setting the unknown flow rate factor to $A = 10^{-16}\text{Pa}^{-3}\text{a}^{-1}$. The spectrum for the approximation errors are shown in Fig. 11. The plot indicates that taking $500 < N \leq 1000$ samples would likely suffice to accurately characterize the approximation errors. For the results discussed here we used $N = 1000$ samples. The average standard deviation of the approximation errors in the $x$-component of the approximation errors is approximately $3.1$, while for

the $y$- and $z$- components, the average standard deviation of the approximation errors are $0.5$ and $0.4$, respectively. The standard deviation of the noise, on the other hand, is $\delta_e \approx 0.25$. We thus can expect the resulting estimates found by disregarding the approximation errors to be unreasonable, see Section 4.1.

In Fig. 12, we show four draws from the prior density on the basal sliding coefficient, while in Fig. 13, we show the true basal sliding coefficient (top left), and each of the MAP estimates; $\beta_{\text{MAP}}^{\text{REF}}$ (top right), $\beta_{\text{MAP}}^{\text{CEM}}$ (bottom left), and $\beta_{\text{MAP}}^{\text{BAE}}$ (bottom right).

We also show the locations ($y = 2.5$km and $y = 7.5$km) of two lines, labeled $l_1$ and $l_2$, for which cross sectional plots are shown in Fig. 14. It is clear from Figures 13 and 14 that the reference posterior is completely feasible, and the corresponding MAP estimate is in good agreement with the true basal sliding coefficient. On the other hand, although the MAP estimate found using the conventional error model (CEM) shows similar qualitatively behavior, as seen in Fig. 13, when taking the corresponding posterior density into account, it is clear that the approach is essentially infeasible, with the truth lying well outside the bulk

of the posterior across most of the domain, see Fig. 14. Finally, from Fig. 13 and 14 we see that, though not as good as the accurate case, the MAP estimate found using the BAE approach qualitatively remains similar to the truth. Furthermore, the truth is generally very well supported by the BAE posterior under the Laplace approximation.

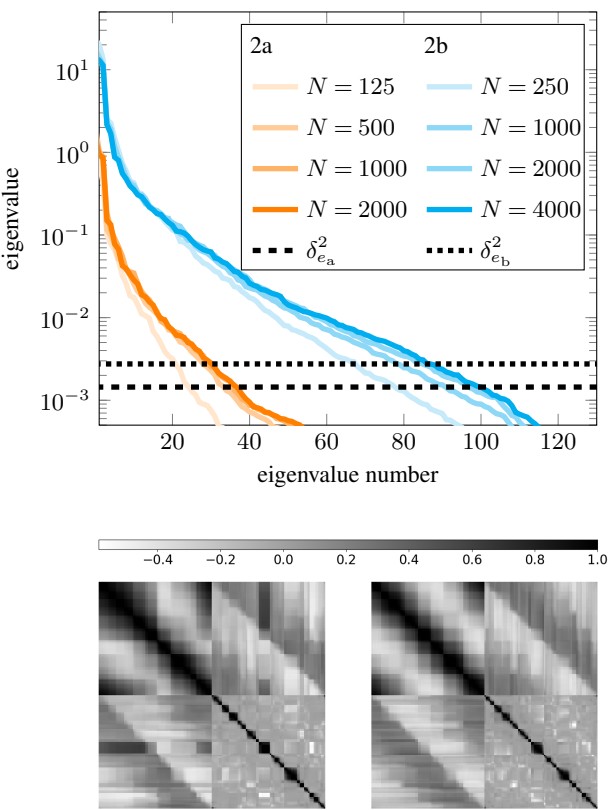

**Figure 9.** Convergence and (Pearson's) correlation matrix of the approximation errors for Example 2. Top row: Spectrum of $\mathbf{\Gamma}_\varepsilon$ for various sample sizes, $N$, for Example 2a (orange) and Example 2b (cyan), along with the noise variance for Example 2a, $\delta^2_{e_\mathrm{a}}$, and 2b, $\delta^2_{e_\mathrm{b}}$. Bottom row: (Pearson's) correlation matrix of the approximation errors for Example 2a (left) and 2b (right).

## 6.4 Spectra of the data misfit Hessians and computational costs.

In this section, we compare the spectra of the data misfit Hessian and the computational cost of the three approaches (accurate, conventional error, and Bayesian approximation error) for each of the three examples. The dominant eigenvalues of the data misfit Hessian, $\overline{\mathbf{H}}$, see (10), evaluated at the corresponding MAP estimate, are shown in Fig. 15 for each example. Firstly, we

5  observe that for all three approaches in all three examples, we only need to retain a relatively low number of eigenvalues and eigenvectors to compute a reasonable low rank approximation of the Laplace posterior covariance matrix. Secondly, we see that the dominant spectrum resulting from using the BAE approach is often lower than that of the reference and conventional error approach cases. This is to be expected since we are accommodating the approximation errors, which naturally lead to an increase in uncertainty. Finally, the dominant spectrum of the misfit Hessian for Examples 1 and 2, found using the conventional

10  error approach, further illustrate the fact that ignoring the uncertainty in the auxiliary parameters can lead to overconfidence in erroneous estimates.

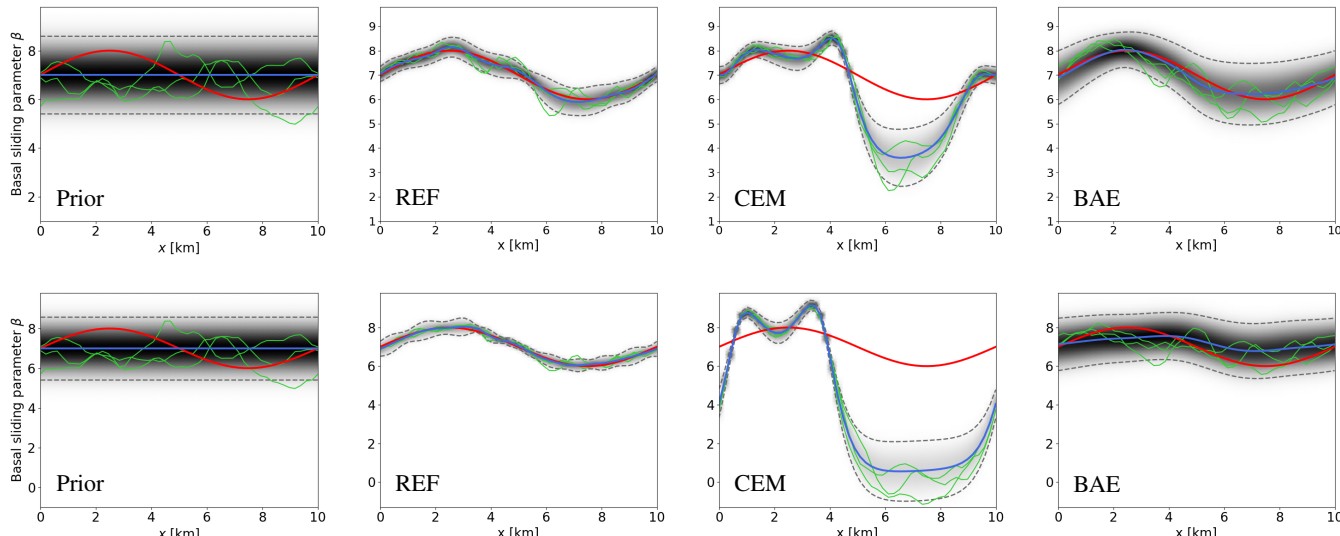

**Figure 10.** Prior and MAP estimates of the basal sliding parameter for Example 2. Top row: Example 2a prior (far left), accurate/reference (REF) case (center left), conventional error model (CEM) case (center right), and Bayesian approximation error (BAE) case (far right). Bottom row shows the same plots for Example 2b. In each plot, the mean of the distribution (blue line) is shown along with three samples from the respective distributions (green lines), the marginal distribution (shaded) with darker shading indicating higher probability, and the $\pm 2$ (approximate) standard deviation intervals (black dashed line).

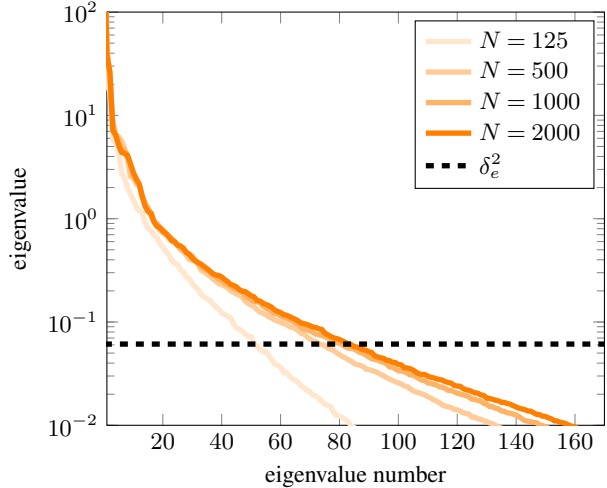

**Figure 11.** Spectrum of $\boldsymbol{\Gamma}_\varepsilon$ for various sample sizes, $N$, for Example 3, along with the noise variance, $\delta_e^2$.

The spectrum of the misfit Hessian for Example 3, found using the conventional error approach, seems to be somewhat anomalous in that the spectrum decays faster than that of the misfit Hessian found using the BAE approach. However, this is possibly explained by the fact that the respective misfit Hessian are evaluated at quite different MAP estimates.





**Figure 12.** Four samples from the prior for the basal sliding parameter field for Example 3.

Figure 15 shows that the number of eigenvalues required to compute a reasonable low rank approximation, in the sense of (12), is considerably lower for the BAE approach in most of the examples. This result suggests that computing the low-rank approximation is cheaper for the BAE approach compared to the other two approaches.

With regard to the computational cost, we consider the number of (linearized) Stokes problem solves required for the optimization algorithm to converge as the unit of cost. As stated in Section 3, we use the inexact Newton-CG algorithm with Armijo line search to find the MAP point. At each iteration, inexact Newton-CG requires:





**Figure 13.** Basal sliding parameter estimates. Top row: The true basal sliding coefficient (left), and the accurate MAP estimate $\beta_{\mathrm{MAP}}^{\mathrm{REF}}$ (right). Bottom row: The conventional error model MAP estimate $\beta_{\mathrm{MAP}}^{\mathrm{CEM}}$ (left), and the Bayesian approximation error MAP estimate $\beta_{\mathrm{MAP}}^{\mathrm{BAE}}$ (right). The black dashed lines are used to show the location of the cross sections ($l_1 = 2.5$km and $l_2 = 7.5$km) for Fig. 14.

a) one (or more if required to satisfy the sufficient descent condition) evaluation of the log-likelihood, which involves solving the nonlinear Stokes equations;

b) one gradient evaluation, which involves solving an additional linearized Stokes problem, i.e., the adjoint equation;


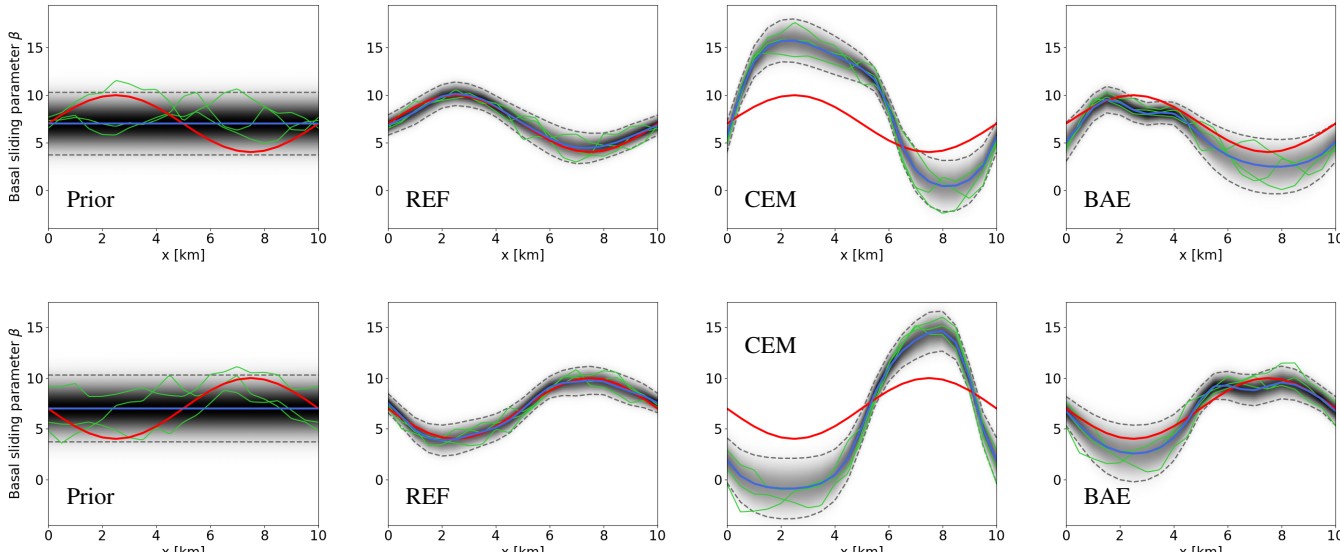

**Figure 14.** Cross-sections of prior and MAP estimates of the basal sliding parameter for Example 3. Top row: Cross section along line $l_1$ ($y = 2.5$km) of prior (far left), accurate/reference (REF) case (center left), conventional error model (CEM) case (center right), and Bayesian approximation error (BAE) case (far right). Bottom row shows cross section along line $l_2$ ($y = 7.5$km) in the same order. In each plot, the mean of the distribution (blue line) is shown along with three samples from the respective distributions (green lines), the marginal distribution (shaded) with darker shading indicating higher probability, and the $\pm2$ (approximate) standard deviation intervals (black dashed line).

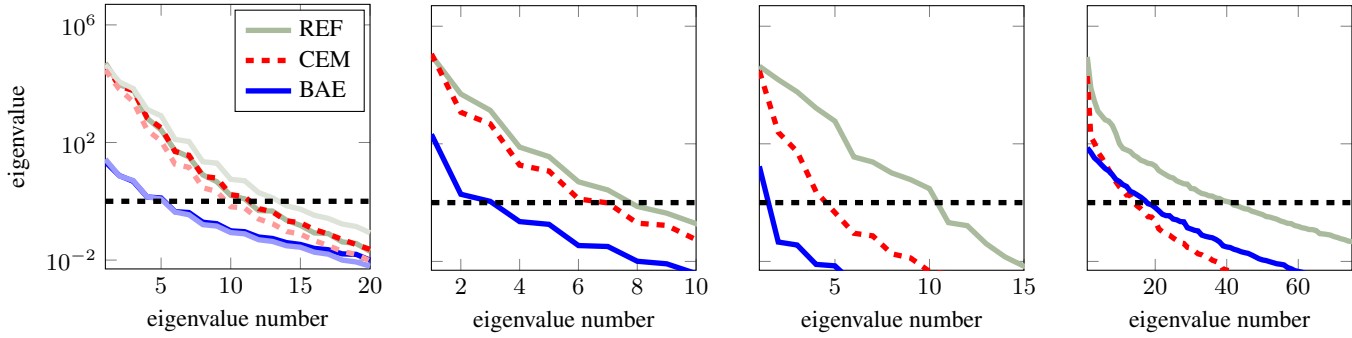

**Figure 15.** Spectra of the prior-preconditioned Hessian of the data misfit computed using (11) for Example 1 (far left), Example 2a (centre left), Example 2b (centre right), and Example 3 (far right). The spectrum for Example 1a (coarse mesh) are shown in the fainter colors, while the spectrum for Example 1b (fine mesh) are shown in the richer colors. The horizontal black dashed-line (at $\lambda = 1$) shows the reference value for the truncation of the spectrum of the prior-preconditioned Hessian of the data misfit.

c) one Newton system solve using the conjugate gradient (CG) method, which at each CG iteration requires solving two linearized Stokes problems, i.e., the incremental forward and adjoint problems.





The total number of linearized Stokes solves required to compute the MAP estimate can then be calculated–per Gauss-Newton iteration–as the sum of the number of iterations required to solve the nonlinear forward problem, plus one adjoint solve (to calculate the gradient), along with one incremental forward solve and one incremental adjoint solve (to calculate the action of the Hessian) per CG iteration. To ensure a sufficient decrease in the objective function at each (Gauss-)Newton iteration, the forward problem may be solved multiple times until the Armijo condition is satisfied, thus further increasing the number of linearized Stokes solves. We omit the derivations of the adjoint, incremental forward, and incremental adjoint equations since these are given in Petra et al. (2012).

The results shown in Table 2 indicate that in each of the examples considered in this paper, the BAE approach generally requires less than half the number of the linearized Stokes solves that are required for the REF case to converge to the MAP point. Furthermore, the conventional error approach requires (in some cases, significantly) more iterations, and thus linearized Stokes solves, than the accurate model. This is to be expected as the optimization is hampered by model mismatch. It is also worth noting that the CEM approach requires substantially more backtracking iterations compared to the REF and BAE approaches, which is inline with Nicholson et al. (2018). Furthermore, the number of CG iterations is significantly reduced for the BAE approach when compared to the CEM and REF case.

## 7 Conclusions

In this paper, we have considered the inference for the basal sliding coefficient field for ice sheet flow problems with uncertain rheology from surface velocity measurements. The rheology parameters of the ice, in particular the flow rate factor and the Glen's flow law exponent, were treated as unknown random fields, which is often the case in reality. We considered examples in both two and three dimensions, and used both the linear and nonlinear Stokes ice sheet model. In each of the cases considered, our goal was to infer the basal sliding coefficient only, as such the unknown rheology parameters were *a priori* fixed to nominal values, and treated as auxiliary parameters. To account for the resulting modeling uncertainties (or errors), we employed the Bayesian Approximation Error (BAE) approach. This approach shifts all uncertainty into a single additive total error term, which is approximated as Gaussian, and can be premarginalized over.

Quantification of the resulting uncertainty in the estimated basal sliding coefficient was carried out based on the Laplace approximation to the posterior. In all of the examples considered here, the results suggest that fixing rheology parameters to standard values found in the literature, can lead to overly confident and (heavily) biased estimates, with the true basal sliding coefficient generally lying outside the bulk of the posterior density, if the uncertainty in the rheology parameters is not accounted for. Conversely, carrying out approximate premarginalization over the unknown rheology parameters, via the BAE approach, leads to feasible estimates for the basal sliding coefficient in all cases considered. To illustrate a limitation of the BAE approach, we included an example in which the modeling errors introduced were, in some sense, *too large*. This case led to a posterior density (found using the BAE approach) which showed very little reduction in variance compared to the prior, though it still contained the truth.

**Table 2.** The cost of solving for the MAP estimates, measured in number of linearized Stokes solves. The first column (Ex.) refers to the example number, and the second column (Est.) refers to which MAP estimate we are solving for, i.e., the reference MAP (REF), the MAP found using the conventional error model (CEM), or the MAP found using the BAE approach (BAE). The third column (#N) gives the number of (Gauss-)Newton iterations, while fourth column (#CG) reports the total number of CG iterations. The fifth column (#back) reports the number of backtracks needed throughout the (Gauss-)Newton iterations, and the sixth column (#O) gives the total number of objective function evaluations. Finally, the last column (#Stokes) gives the total number of linearized Stokes solves (for forward, adjoint, incremental forward, and incremental adjoint problems). The (Gauss-)Newton iterations are terminated when the norm of the gradient is decreased by a factor of $10^6$, while the CG iterations are terminated inline with the Eisenstat-Walker condition (Eisenstat and Walker, 1996) (to avoid over-solving) and the Steihaug criteria (Steihaug, 1983) (to avoid negative curvature). The results illustrate that the use of the approximation error approach can be carried out at no additional online cost compared to the conventional error approach and reference case.

| Ex. | Est. | #O | #N | #CG | #back | #Stokes |
|---|---|---|---|---|---|---|
| | REF | 10 | 10 | 68 | 0 | 156 |
| 1a | CEM | 19 | 17 | 96 | 2 | 228 |
| | BAE | 7 | 7 | 30 | 0 | 74 |
| | REF | 12 | 11 | 68 | 1 | 159 |
| 1b | CEM | 18 | 16 | 91 | 2 | 216 |
| | BAE | 7 | 7 | 28 | 0 | 70 |
| | REF | 21 | 16 | 66 | 5 | 244 |
| 2a | CEM | 42 | 27 | 99 | 15 | 376 |
| | BAE | 12 | 11 | 25 | 1 | 110 |
| | REF | 12 | 11 | 78 | 1 | 241 |
| 2b | CEM | 35 | 26 | 62 | 9 | 286 |
| | BAE | 8 | 8 | 18 | 0 | 78 |
| | REF | 18 | 16 | 178 | 2 | 491 |
| 3 | CEM | 37 | 23 | 115 | 14 | 438 |
| | BAE | 14 | 13 | 64 | 1 | 240 |



By avoiding simultaneous estimation of the basal sliding coefficient and rheology parameters (which are spatially varying over the entire domain) the online computational overheads of the estimation problem are substantially reduced. To ensure the work carried out here is applicable to large-scale problems, i.e., scalable, we initially posed the problem in infinite dimensions and then employed the adjoint-state methodology to compute the MAP estimate.

5    In assessing the applicability and performance of the BAE approach, the current study only considers fairly limited domains, i.e., box-like geometries, and idealized boundary conditions. A natural next step for future work is to apply the same framework to more realistic setups and to continental-scale ice flow problems.

### 7.1

*Competing interests.*  No competing interests are present.

10    *Acknowledgements.*  This work was supported by the U.S. National Science Foundation, Software Infrastructure for Sustained Innovation (SI2: SSE & SSI) Program under grants ACI-1550593, and ACI-1550547 and by NSF Division of Mathematical Sciences under the CAREER grant 1654311. The authors gratefully acknowledge computing time on the Multi-Environment Computer for Exploration and Discovery (MERCED) cluster at UC Merced, which was funded by National Science Foundation Grant No. ACI-1429783.





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
