# Peer review of "Inferring the basal sliding coefficient field for the Stokes ice sheet model under rheological uncertainty"

_The Cryosphere, 2020_

## Referee Comment (RC1) · Douglas Brinkerhoff (Referee) · 5 Oct 2020

In *Inferring the basal sliding coefficient field for the Stokes ice sheet model under rheological uncertainty*, Babaniyi and co-authors present the application of a Laplace-approximation based Bayesian inference procedure to the problem of determining the distribution of basal traction given observations. In particular, the authors make a significant contribution to inverse problems in glaciology by demonstrating what happens when we ignore the uncertainties in model parameters apart from the ones that we're primarily interested in. In particular they show that ignorming a reasonable amount of rheological uncertainty leads to a posterior distribution that is egregiously incorrect and

far too sure of itself. They address this problem by introducing the Bayesian Approximation Error approach, in which random samples from the nuisance variable are used to generate an empirical covariance and bias in the model predictions, which are added to observational noise to produce a much more realistic pseudo-likelihood model and thus a much more reasonable posterior distribution.

This paper is an excellent and timely contribution. First, it builds upon a lineage of important papers (Bui-Tanh, 2013; Petra, 2014; Isaac, 2015) that have provided a framework for (Bayesian) uncertainty quantification in large-scale ice sheet model (something that is of critical importance, in my view), while also demonstrating the limitations of those methods that still remain. In some ways, this paper raises more questions than it answers (and I mean that in a good way): if even a relatively robust inversion framework like Bayesian inference can be led astray by unquantified aleatoric model uncertainty, how can we going to deal with unquantified epistemic uncertainty going forward? Second, numerical recipe that it introduces is general and viable across a large class of problems in ice sheet modelling, so long as there are a sufficient number of cores available for drawing Monte Carlo samples. The development of "embarassingly parallel" methods for uncertainty quantification (as opposed to ones that require more intricate communication) make the implementation of these methods broadly feasible for other resesearchers. Finally, the results are compelling. The posterior distributions that the authors produce are indeed much more realistic than those produced in the absence of their method, and any work that makes the case for a *broader* posterior variance in ice sheet model inversions should be lauded.

Nonetheless, There are a handful of factors that may limit the paper's applicability. I describe these issues below, interspersed with a handful of technical corrections.

**P6 L15** The choice of covariance function should be motivated. First, why bother with this inverse elliptic operation to begin with, when it doesn't seem any more physical in this case than using squared exponential or Matern kernel? If elliptic inverse is justified, then why the exponent of 2? This differs from that used by Petra (2014), so why the change?

**P7 L10** Fix reference formatting.

**P8 L7** It would be better to use Isaac (2015a) as the reference here, since the "generalized eigenvalue problem" formulation does not appear in the previous works (although the solution to the problem stated in those works is equivalent).

**P8 L12** What does *sufficiently* mean?

**Eq. 14, 16** This 'right arrow' notation isn't very informative.

**P9 L13** $\epsilon_*$ is defined (empirically) later, but it should at least be given a name or description here.

**Eq. 21** This method seems to rely on the modelling error being well-approximated by a normal distribution. Is there any evidence to suggest that this is true for the problems being considered here? (The forward model is non-linear after all). What happens if it's not?

**P10 L11–18** These 'rules of thumb' need some explanation. What are they telling us and why are they different? A qualitative description would help.

**P11 L15–19** This is a very unrealistic prior relative to the typical situation in ice sheet modelling. In what case do we typically know the true value of a parameter's mean? That's a lot of information! I think a much more interesting case would be to provide a vague prior with a mean not equal to the true value. A frequent assumption for GRFs is mean zero and a large marginal variance, because there isn't any more information than that to go off of *a priori*. Does this method still work for a less informative prior, or does it ruin the estimates of $\epsilon_*$ and $\Gamma_\epsilon$?

**P13 L1** Why introduce rejection sampling when one could use a similar log-transform as for the enhancement factor? Why does constraining the parameter space to shear thinning matter in this synthetic case anyways?

**P13 L13** Where does the weird constant 3 come from in the rate factor parameterization?

**Table 1** These values on prior distributions seem arbitrary, and because they're parameters of an elliptic model rather than the more interpretable length scale and variance that we'd find in a more traditional GRF kernel, they are a bit opaque. Why were these values of $\gamma$. and $\delta$. chosen?

**Sections 5/6** I think that these sections should be reordered to put both setup and results for each example right next to one another. This would be more brief, and the reader wouldn't have to flip back and forth between sections.

**Figure 5 and supporting text** It's unusual for vertical velocity to be available in an inversion (we don't really get it from remotely sensed data). Would it change results much to not have access to it?

**P17 L6** 'Finally, we see that the off-diagonal blocks also display fairly complex behavior' is not a useful sentence. Complex in what way, and why?

**P27 L6–7** 'We omit...' This sentence is redundant.
* * *

---

## Referee Comment (RC2) · Anonymous Referee #2 · 17 Oct 2020

The paper "Inferring the basal sliding coefficient field for the Stokes ice sheet model under rheological uncertaint" by Babaniyi and co-authors, addresses the challenging topic of accounting for modeling errors when estimating the basal sliding coefficient $\beta$ from surface velocity observations. In particular, the paper considers uncertainty in the rehology and demonstrate how it can be properly accounted for using the Bayesian Approximation Error approach. Several numerical results on simplified ISMIP-HOM like problems demonstrate the effectiveness of their approach and the failure of the traditional approach. I think the paper is well written and it represent a significant contribution to the ice sheet community. Before recommending it for publication, I would like the authors to address the following concerns:

[Figure]

1. While I am convinced of the effectiveness and usefulness of the proposed approach, I am wondering whether the difference between the proposed approach and the traditional one has been a overemphasized by taking a regularization (prior) for $\beta$ that is too small. In fact, it seems to me that the data are over-fitted when using the traditional approach (REF/CEM). It would be interesting to see what happens when $\gamma_\beta$ and $\delta_\beta$ are increased (one could do that using the the L-curve rule, for the deterministic inversion to compute the MAP point). In general, I think that the parameters used for all the priors should be motivated.

2. In real applications, the flow factor is not considered uniform, but it is a function of the temperature. Of course, because of modeling errors, the rheology will still be affected by uncertainty. I'm wondering how your approach and results would change if the parameter $a^*$ were spatially variable.

3. I think it would be better if the true value of the parameter $a_{true}$ were not sampled from the same distribution used for computing the statistics for the approximation error, but from another distribution (e.g. non Gaussian/with different mean/variance)? As a matter of fact, we don't know the distribution for $a$.

4. The authors make the point that the (offline) computation of the statistics for the approximation error requires a "fairly small number of samples". This is true for the numerical results reported in the paper. However, I would argue that in real applications, with complex geometries and real data, that won't necessarily be the case. As a different but related example, the number of eigenvalues needed to accurately approximate the prior-preconditioned Hessian for beta, in the examples reported in this paper, is about two order of magnitude smaller than that needed for the Antarctic ice sheet (Isaac et al. SISC, 2015).

Minor comments:

abstract: I would specify, both in the abstract and in the introduction that the paper is targeting synthetic/manufactured geometries and data.

p. 8, l. 24: Please motivate the assumption of Gaussianity of the unknown parameters. I think there is little evidence to suggest that the distribution of these parameters is Gaussian, and anyway in general we do not know parameters such $\gamma$ and $\delta$. Is the Gaussianity required by the Bayesian Approximation Error theory?

p. 8, l. 28 : Can you please explicitly write (not necessarily here) how the samples are computed using the covariance matrix? think the readers of this journal could benefit from that.

Table 1: Report units of $\delta$ and $\gamma$. Also, in my understanding $\sqrt{\frac{\gamma}{\delta}}$ is a correlation length for the samples. It might be worth pointing that out.

sect. 5.1: In each of the 3 examples, I would remind the reader that $a_{true}$ is chosen as previously shown in Figures 3 and 4.

p. 27, l. 17: I would rephrase this sentence"The rheology parameters of the ice, in particular the flow rate factor and the Glen's flow law exponent, were treated as unknown random fields, which is often the case in reality." In real application the rehology is computed out of the temperature based on physical models.

---

## Author Comment (AC2) · 1 Dec 2020

Thanks for the careful reading and for your helpful comments and suggestions. Please find attached point-by-point replies (in black) to your comments and questions (which are reprinted in blue). To give you an overview of all the changes in the paper, we also provide a diff-document that highlights the changes between the initial submission and this re-submission.

Please also note the supplement to this comment:
https://tc.copernicus.org/preprints/tc-2020-229/tc-2020-229-AC2-supplement.pdf

---

## Author Response (AR1)

**Reply to the Reviewer 1:**

Thanks for the careful reading and for your helpful comments and suggestions. Please find below point-by-point replies (in black) to your comments and questions (which are reprinted in blue). To give you an overview of all the changes in the paper, we also provide a diff-document that highlights the changes between the initial submission and this re-submission.

In inferring the basal sliding coefficient field for the Stokes ice sheet model under rheological uncertainty, Babaniyi and co-authors present the application of a Laplace approximation based Bayesian inference procedure to the problem of determining the distribution of basal traction given observations. In particular, the authors make a significant contribution to inverse problems in glaciology by demonstrating what happens when we ignore the uncertainties in model parameters apart from the ones that were primarily interested in. In particular they show that ignoring a reasonable amount of rheological uncertainty leads to a posterior distribution that is egregiously incorrect and far too sure of itself. They address this problem by introducing the Bayesian Approximation Error approach, in which random samples from the nuisance variable are used to generate an empirical covariance and bias in the model predictions, which are added to observational noise to produce a much more realistic pseudo-likelihood model and thus a much more reasonable posterior distribution. This paper is an excellent and timely contribution. First, it builds upon a lineage of important papers (Bui-Tanh, 2013; Petra, 2014; Isaac, 2015) that have provided a framework for (Bayesian) uncertainty quantification in large-scale ice sheet model (something that is of critical importance, in my view), while also demonstrating the limitations of those methods that still remain. In some ways, this paper raises more questions than it answers (and I mean that in a good way): if even a relatively robust inversion framework like Bayesian inference can be led astray by unquantified aleatoric model uncertainty, how can we going to deal with unquantified epistemic uncertainty going forward? Second, numerical recipe that it introduces is general and viable across a large class of problems in ice sheet modelling, so long as there are a sufficient number of cores available for drawing Monte Carlo samples. The development of "embarassingly parallel" methods for uncertainty quantification (as opposed to ones that require more intricate communication) make the implementation of these methods broadly feasible for other researchers. Finally, the results are compelling. The posterior distributions that the authors produce are indeed much more realistic than those produced in the absence of their method, and any work that makes the case for a broader posterior variance in ice sheet model inversions should be lauded. Nonetheless, There are a handful of factors that may limit the papers applicability. I describe these issues below, interspersed with a handful of technical corrections.

1. **P6 L15** The choice of covariance function should be motivated. First, why bother with this inverse elliptic operator to begin with, when it doesnt seem any more physical in this case than using squared exponential or Matern kernel? If elliptic inverse is justified, then why the exponent of 2? This differs from that used by Petra (2014), so why the change?

   We agree with the reviewer that some additional motivation for the prior should be given, and we have now incorporated more discussion about the choice of the prior in the manuscript, please see Section 3.1. It has been shown (see e.g. [7, 15] and references in the revised manuscript) that the prior we use is equivalent to a Matérn prior with a certain smoothness parameter which depends on the number of space dimensions. In particular for the three-dimensional problem in example 3, our prior covariance is equivalent to that exponential covariance function. However, defining the prior covariance as the inverse of an elliptic operator has several computational advantages since evaluating the prior distribution only requires a matrix-vector product with a sparse precision matrix, and sampling only requires solving a symmetric positive definite linear system, for which

fast and scalable multigrid methods can be employed. In [12], the authors define the field $m$ on the boundary; here due to some technical limitations of FEniCS, we had to define the covariance on the whole domain and then restrict it to the boundary.

2. **P7 L10** Fix reference formatting.

   Thank you for pointing this out. In the revised manuscript we have fixed this issue.

3. **P8 L7** It would be better to use Isaac (2015a) as the reference here, since the "generalized eigenvalue problem" formulation does not appear in the previous works (although the solution to the problem stated in those works is equivalent).

   We have now updated the manuscript to reference [2].

4. **P8 L12** What does sufficiently mean?

   We agree, this is indeed quite vague. In the revised manuscript, we have incorporated more precise guidelines on how the truncation index is chosen along with references. Furthermore, we have corrected a typo in the generalized eigenvalue problem. Below is a slightly more detailed explanation (which we believe is best left out of the current manuscript) which can be found in, for example, [2]. We start with the full posterior covariance matrix, $\mathbf{\Gamma}_{\mathrm{po}} \in \mathbb{R}^{m \times m}$, which can be written as

$$\mathbf{\Gamma}_{\mathrm{po}} = \mathbf{\Gamma}_{\mathrm{pr}} - \mathbf{V}_r \mathbf{D}_r \mathbf{V}_r^T + \mathcal{O}\left( \sum_{i=r+1}^{n} \frac{\lambda_i}{\lambda_i + 1} \right),$$

   for $\mathbf{D}_r = \mathbf{\Lambda}_r (\mathbf{I}_r + \mathbf{\Lambda}_r)^{-1} \in \mathbb{R}^{m \times m}$ and $\mathbf{\Lambda}_r \in \mathbb{R}^{m \times m}$ and $\mathbf{V}_r \in \mathbb{R}^{m \times r}$ satisfying the generalized eigenvalue problem

$$\overline{\mathbf{H}} \mathbf{V}_r = \mathbf{\Gamma}_{\mathrm{pr}}^{-1} \mathbf{V}_r \mathbf{\Lambda}_r.$$

   Here $\overline{\mathbf{H}}$ is the Hessian of the data misfit component of negative log-posterior. We choose $r$ such that

$$\mathcal{O}\left( \sum_{i=r+1}^{n} \frac{\lambda_i}{\lambda_i + 1} \right)$$

   is small, which happens for $\lambda_i \ll 1$ (in practice we choose $r$ when $\lambda_i$ is about 0.01). Once $r$ is determined, the approximation of the posterior covariance can be written as

$$\mathbf{\Gamma}_{\mathrm{po}} \approx \mathbf{\Gamma}_{\mathrm{pr}} - \mathbf{V}_r \mathbf{D}_r \mathbf{V}_r^T.$$

5. **Eq. 14, 16** This right arrow notation isnt very informative.

   We have changed this in the revised manuscript.

6. **P9 L13** $\epsilon_*$ is defined (empirically) later, but it should at least be given a name or description here.
   We have now introduced $\varepsilon_*$ and $\mathbf{\Gamma}_\varepsilon$ as the mean and covariance of the approximation errors.

7. **Eq. 21** This method seems to rely on the modelling error being well-approximated by a normal distribution. Is there any evidence to suggest that this is true for the problems being considered here? (The forward model is non-linear after all). What happens if its not?

The standard BAE approach does take a normal approximation to the modeling errors, by considering only the mean and covariance matrix, however, as the total errors are premarginalized over, it is in fact the effect of this marginalization we hope to capture. There is empirical evidence that this Gaussian approximation is a sensible choice, see for example[3, 13, 6, 1, 9, 11]. Gaussian process approximation to model error was also advocated in [4]. Testing for normality could be done visually or more formally using, for example, a ShapiroWilk test.

We also point out that some recent work has been carried out in which the normality approximation has been dropped [8, 10, 16]. These approaches, however, are generally based on machine learning techniques, such as random forests [8], convolutional neural networks [10], or deep neural networks [16], which come with their own set of (computational and otherwise) challenges.

8. **P10 L1118** These rules of thumb need some explanation. What are they telling us and why are they different? A qualitative description would help.

The *rules of thumb* provide a guideline for determining when the approximation *dominates* the measurement noise. We have updated this paragraph to included more intuition.

9. **P11 L1519** This is a very unrealistic prior relative to the typical situation in ice sheet modelling. In what case do we typically know the true value of a parameters mean? Thats a lot of information! I think a much more interesting case would be to provide a vague prior with a mean not equal to the true value. A frequent assumption for GRFs is mean zero and a large marginal variance, because there isnt any more information than that to go off of a priori. Does this method still work for a less informative prior, or does it ruin the estimates of $\epsilon_*$ and $\Gamma_\epsilon$

The reviewer is right, we typically don't know much a-priori about the basal boundary coefficient field. However, one can argue that the topography and especially the top surface velocity field can give us some prior information about the basal boundary condition and one could construct a useful prior (for instance we know that we cannot reconstruct highly oscillatory features of the basal boundary conditions, hence the samples from the prior should be fairly smooth, etc.). The mean value $\beta_*$ of the prior here is the same as the spatial mean of the $\beta$ by coincidence. Our conclusion does not change if we change the prior mean and variance (as long as it makes sense physically). Though as expected, a more variable prior will generally lead to larger uncertainty, and a true $\beta$ that is less well supported by the prior will likely not be estimated as well. We have run a couple of simulations for Example 1 to illustrate this. In Figure 1, we show results obtained with prior mean $\beta_* = 6$ (first row), $\beta_* = 5$ (second row) and $\beta_* = 5$ plus increased variance (bottom row). These results show that the conclusion indeed does not change when using different prior information.

[Figure]

Figure 1: Use of different priors on $\boldsymbol{\beta}$ and resulting posteriors for Example 1a (of the manuscript). Top row: Prior mean taken as $\boldsymbol{\beta}_* = 6$ and variance unchanged, Middle row: Prior mean taken as $\boldsymbol{\beta}_* = 5$ and variance unchanged, Bottom row: Prior mean taken as $\boldsymbol{\beta}_* = 5$ and variance increased.

10. **P13 L1** Why introduce rejection sampling when one could use a similar log-transform as for the enhancement factor? Why does constraining the parameter space to shear thinning matter in this synthetic case anyways?

    Various transforms such as the $\log$-transform could be used, and we updated the manuscript to acknowledge this. The rejection sampling approach has the benefit of truncating the distribution, while otherwise leaving the distribution unchanged. On the other hand, the samples are rejected before any simulations are carried out.

11. **P13 L13** Where does the weird constant 3 come from in the rate factor parameterization?

    Thanks for pointing this out. We agree that this is confusing as the constant was originally 1 in our parameterization of the flow rate factor for Example 1 (the linear Stokes case). This constant is there to make sure that the effective viscosity simplifies to $\eta(\boldsymbol{u}) = \frac{1}{2} \exp(a(\boldsymbol{x})) A_0^{-\frac{1}{n}} \dot{\boldsymbol{\varepsilon}}_{\mathrm{II}}^{-\frac{1-n}{2n}}$ meaning that the constant has to equal to the Glen's flow law exponent. We have redefined our parameterization of the flow rate factor in the paper to be $A = A_0 \exp(-na)$ to make it clear that it always depends on $n$.

12. **Table 1** These values on prior distributions seem arbitrary, and because theyre parameters of an elliptic model rather than the more interpretable length scale and variance that wed find in a more

traditional GRF kernel, they are a bit opaque. Why were these values of $\gamma_{\cdot}$ and $\delta_{\cdot}$ chosen?

The parameters in the elliptic PDE-based prior have a nice physical interpretation as well. In particular, $\gamma_\beta$ and $\delta_\beta$ control the correlation length and the marginal variance of the prior distribution. Specifically, the correlation length (defined as the distance for which the two-points have a correlation coefficient of 0.1) is proportional to $\sqrt{\gamma_\beta/\delta_\beta}$, while the variance is proportional to $\delta_\beta^{-2}\left(\gamma_\beta/\delta_\beta\right)^{\frac{d-1}{2}}$, see for example [5] and the references therein. The values for these parameters were chosen by visually inspecting samples from the distributions. We have added these details to the manuscript.

13. **Sections 5/6** I think that these sections should be reordered to put both setup and results for each example right next to one another. This would be more brief, and the reader wouldnt have to flip back and forth between sections.

We understand this concern. However, we believe that the current setup allows readers to focus on the results rather then having to read through a long section mixed with the problem setup, and perhaps more importantly this setup allows us to create a summary table to which we can refer from the results sections easily. For these reasons, we prefer to keep the current setup.

14. **Figure 5** and supporting text Its unusual for vertical velocity to be available in an inversion (we dont really get it from remotely sensed data). Would it change results much to not have access to it?

We agree that the vertical velocities are not always available. However, as shown in [14], they are also usually fairly non-informative. Furthermore, the assumed noise level in the current paper is larger than the vertical velocities, and we thus expect the results would not change if we ignored the vertical velocity data.

15. **P17 L6** 'Finally, we see that the off-diagonal blocks also display fairly complex behavior' is not a useful sentence. Complex in what way, and why?

We have revised this statement to be more precise.

16. **P27 L67** We omit... This sentence is redundant.

We agree, and have modified the manuscript accordingly.

**References**

[1] T. J. Evans, R. Harper, and S. T. Flammia, *Scalable Bayesian Hamiltonian learning*, arXiv preprint arXiv:1912.07636, (2019).

[2] T. Isaac, N. Petra, G. Stadler, and O. Ghattas, *Scalable and efficient algorithms for the propagation of uncertainty from data through inference to prediction for large-scale problems, with application to flow of the Antarctic ice sheet*, Journal of Computational Physics, 296 (2015), pp. 348–368.

[3] J. Kaipio and V. Kolehmainen, *Bayesian Theory and Applications*, Oxford University Press, 2013, ch. Approximate Marginalization Over Modeling Errors and Uncertainties in Inverse Problems, pp. 644–672.

[4] M. C. Kennedy and A. O'Hagan, *Bayesian calibration of computer models*, Journal of the Royal Statistical Society: Series B (Statistical Methodology), 63 (2001), pp. 425–464.

[5] U. Khristenko, L. Scarabosio, P. Swierczynski, E. Ullmann, and B. Wohlmuth, *Analysis of boundary effects on PDE-based sampling of Whittle–Matérn random fields*, SIAM/ASA Journal on Uncertainty Quantification, 7 (2019), pp. 948–974.

[6] T. Lähivaara, N. Dudley Ward, T. Huttunen, Z. Rawlinson, and J. Kaipio, *Estimation of aquifer dimensions from passive seismic signals in the presence of material and source uncertainties*, Geophysical Journal International, 200 (2015), pp. 1662–1675.

[7] F. Lindgren, H. Rue, and J. Lindström, *An explicit link between Gaussian fields and Gaussian Markov random fields: The stochastic partial differential equation approach*, Journal of the Royal Statistical Society: Series B (Statistical Methodology), 73 (2011), pp. 423–498.

[8] A. Lipponen, J. M. Huttunen, S. Romakkaniemi, H. Kokkola, and V. Kolehmainen, *Correction of model reduction errors in simulations*, SIAM Journal on Scientific Computing, 40 (2018), pp. B305–B327.

[9] A. Lipponen, T. Mielonen, M. R. Pitkänen, R. C. Levy, V. R. Sawyer, S. Romakkaniemi, V. Kolehmainen, and A. Arola, *Bayesian aerosol retrieval algorithm for MODIS AOD retrieval over land*, Atmospheric Measurement Techniques, 11 (2018), pp. 1529–1529.

[10] S. Lunz, A. Hauptmann, T. Tarvainen, C.-B. Schönlieb, and S. Arridge, *On learned operator correction*, arXiv preprint arXiv:2005.07069, (2020).

[11] K. Muhumuza, J. M. Huttunen, L. Roininen, and T. Lahivaara, *A Bayesian-based approach to improving acoustic Born waveform inversion of seismic data for viscoelastic media*, Inverse Problems, (2020).

[12] N. Petra, J. Martin, G. Stadler, and O. Ghattas, *A computational framework for infinite-dimensional Bayesian inverse problems: Part II. Stochastic Newton MCMC with application to ice sheet inverse problems*, SIAM Journal on Scientific Computing, 36 (2014), pp. A1525–A1555.

[13] A. Pulkkinen, V. Kolehmainen, J. P. Kaipio, B. T. Cox, S. R. Arridge, and T. Tarvainen, *Approximate marginalization of unknown scattering in quantitative photoacoustic tomography*, Inverse Problems & Imaging, 8 (2014), p. 811.

[14] M. J. Raymond and H. Gudmundsson, *Estimating basal properties of ice streams from surface measurements: a non-linear Bayesian inverse approach applied to synthetic data*, The Cryosphere, 3 (2009), pp. 265–278.

[15] L. Roininen, J. M. J. Huttunen, and S. Lasanen, *Whittle-matérn priors for Bayesian statistical inversion with applications in electrical impedance tomography*, Inverse Problems & Imaging, 8 (2014), p. 561.

[16] S. Sheriffdeen, J. C. Ragusa, J. E. Morel, M. L. Adams, and T. Bui-Thanh, *Accelerating pde-constrained inverse solutions with deep learning and reduced order models*, arXiv preprint arXiv:1912.08864, (2019).

**Reply to the Reviewer 2:**

Thanks for the careful reading and for your helpful comments and suggestions. Please find below point-by-point replies (in black) to your comments and questions (which are reprinted in blue). To give you an overview of all the changes in the paper, we also provide a diff-document that highlights the changes between the initial submission and this re-submission.

The paper "inferring the basal sliding coefficient field for the Stokes ice sheet model under rheological uncertainty" by Babaniyi and co-authors, addresses the challenging topic of accounting for modeling errors when estimating the basal sliding coefficient $\beta$ from surface velocity observations. In particular, the paper considers uncertainty in the rheology and demonstrate how it can be properly accounted for using the Bayesian Approximation Error approach. Several numerical results on simplified ISMIP-HOM like problems demonstrate the effectiveness of their approach and the failure of the traditional approach. I think the paper is well written and it represent a significant contribution to the ice sheet community. Before recommending it for publication, I would like the authors to address the following concerns:

1. While I am convinced of the effectiveness and usefulness of the proposed approach, I am wondering whether the difference between the proposed approach and the traditional one has been overemphasized by taking a regularization (prior) for $\beta$ that is too small. In fact, it seems to me that the data are overfitted when using the traditional approach (REF/CEM). It would be interesting to see what happens when $\gamma_\beta$ and $\delta_\beta$ are increased (one could do that using the L-curve rule, for the deterministic inversion to compute the MAP point). In general, I think that the parameters used for all the priors should be motivated.

   It would be extremely difficult (likely impossible), to take into account the correlation structure embedded in (the covariance matrix of) the approximation errors via a regularization parameter, thus loosing valuable information. Furthermore, the mean of the approximation errors is not *negligible* in all cases, that is, disregarding the unknown rheological parameters induces systematic bias, which would be challenging (at best) to accomodate by changing $\gamma_\beta$ and $\delta_\beta$.

2. In real applications, the flow factor is not considered uniform, but it is a function of the temperature. Of course, because of modeling errors, the rheology will still be affected by uncertainty. I'm wondering how your approach and results would change if the parameter $a_*$ were spatially variable.

   Indeed the flow factor is in practice not constant, and is generally temperature dependent. The choice of $a_*$ was chosen only to be in line with commonly chosen values in the literature. The prior chosen for $a$ in each of the examples is based on the assumption that we expect smooth variations of $a$ about the mean value, $a_*$. If on the other hand we had (a priori) reason to believe that $a$ smoothly varied about some other (nonuniform) value, this could certainly be incorporated. In such a case, we expect the results would be qualitatively the same.

3. I think it would be better if the true value of the parameter $a_{\text{true}}$ were not sampled from the same distribution used for computing the statistics for the approximation error, but from another distribution (e.g. non Gaussian/ with different mean/variance)? As a matter of fact, we don't know the distribution for $a$.

   We agree that $a_{\text{true}}$ could be sampled from a different distribution to fully demonstrate the robustness of the BAE approach in situations where $a_{\text{true}}$ or its distribution is unknown. The choice of $a_{\text{true}}$ used in the manuscript was based on ease of exposition, and convenience, however

[Figure]

[Figure]

[Figure]

[Figure]

Figure 1: Prior and MAP estimates of the basal sliding parameter for Example 1a with the true auxiliary parameter set to $a_{\text{true}} = \sin(2\pi x_1/L) + \sin(2\pi x_2/H)$. Prior (far left), accurate/reference (REF) case (centre left), conventional error model (CEM) case (center right), and Bayesian approximation error (BAE) case (far right). The "truth" (red line), the prior mean and the MAP point (blue line) in the left plot and the right three plots, respectively are shown along with three samples from the respective distributions (green lines), the marginal distribution under Laplace approximation (shaded) with darker shading indicating higher probability, and the $\pm 2$ (approximate) standard deviation intervals (black dashed line).

we expect the approach to perform similarly for a different $a_{\text{true}}$ (not sampled from the prior used to generate the statistics of the approximation errors, but still supported by the prior). To test this hypothesis, in Fig. 1 we show results for Example 1a, where $a_{\text{true}}$ was defined as a sinusoidal function, namely $a_{\text{true}} = \sin(2\pi x_1/L) + \sin(2\pi x_2/H)$. The results reveal similar conclusions.

4. The authors make the point that the (offline) computation of the statistics for the approximation error requires a "fairly small number of samples". This is true for the numerical results reported in the paper. However, I would argue that in real applications, with complex geometries and real data, that won't necessarily be the case. As a different but related example, the number of eigenvalues needed to accurately approximate the prior-preconditioned Hessian for data, in the examples reported in this paper, is about two order of magnitude smaller than that needed for the Antarctic ice sheet (Isaac et al. SISC, 2015).

The number of eigenvalues and eigenvectors of the prior-preconditioned data misfit Hessian that must be retained to give an accurate approximation for the posterior covariance matrix only provides an upper bound on the rank of the approximation error covariance matrix. The rank of the approximation error covariance matrix (and thus the number of samples required) is dictated by how accurate/inaccurate the approximative model, $g(\beta)$, is relative to the accurate model, $f(\beta, a)$. That is, the more accurate the approximative model is, the less samples we require to characterize the covariance of the approximation errors. This also helps to motivate the use of a physically based approximative models (rather than a completely arbitrary model). Therefore, for the large-scale problem defined on the Pine Island or Antartica geometries, we expect the rank of the model error covariance–and therefore the number of samples needed to compute the BAE statistics–to be much smaller than the number of informed eigen-directions of the Hessian of the negative log-likelihood.

**Minor comments:**

- **abstract:** I would specify, both in the abstract and in the introduction, that the paper is targeting synthetic/manufactured geometries and data

Thank you for this suggestion, we have changed this in the revised manuscript.

- **p. 8, l. 24:** Please motivate the assumption of Gaussianity of the unknown parameters. I think there is little evidence to suggest that the distribution of these parameters is Gaussian, and anyway in general we do not know parameters such as $\gamma$ and $\delta$. Is the Gaussianity required by the Bayesian Approximation Error theory?

  The choice of a Gaussian prior distribution is in line with several related previous works carried out within the Bayesian framework, see, for example [1, 6, 8, 7]. Also, postulating a Gaussian prior leads to an optimization problem which is fairly similar to that solved in the deterministic framework using Tikhonov-type regularization, and is thus related to e.g., [5, 2, 4].

  As discussed below, prescribing values for $\gamma$ and $\delta$ is equivalent to providing a prior variance and correlation length, which we believe is not overly restrictive, and is fairly standard for Bayesian inference of distributed parameters in PDE forward models.

  The Bayesian approximation error approach does not assume that any prior distribution is Gaussian, see e.g., [3]. It only requires the ability of sampling from the prior distribution and of evaluating the log-distrubution.

- **p. 8, l. 28:** Can you please explicitly write (not necessarily here) how the samples are computed using the covariance matrix? I think the readers of this journal could benefit from that.

  To sample from a Gaussian distribution, $\mathcal{N}(\boldsymbol{\beta}_*, \boldsymbol{\Gamma}_\beta)$ with $\boldsymbol{\beta}_* \in \mathbb{R}^m$ and $\boldsymbol{\Gamma}_\beta \in \mathbb{R}^{m \times m}$, we require a matrix, $\boldsymbol{L} \in \mathbb{R}^{p \times m}$ ($p \geq m$) such that $\boldsymbol{L}\boldsymbol{L}^T = \boldsymbol{\Gamma}_\beta$. With such a matrix in hand (common choices for $\boldsymbol{L}$ are the Cholesky-factor, or the principal square root), a sample, $\boldsymbol{s} \in \mathbb{R}^m$, from $\mathcal{N}(\boldsymbol{\beta}_*, \boldsymbol{\Gamma}_\beta)$ can be generated taking

  $$\boldsymbol{s} = \boldsymbol{\beta}_* + \boldsymbol{L}\boldsymbol{\eta},$$

  where $\boldsymbol{\eta}$ is sampled from the so-called *standard Gaussian*, i.e., $\boldsymbol{\eta} \sim \mathcal{N}(\boldsymbol{0}, \boldsymbol{I})$. To verify, note firstly, that $\mathbb{E}[\boldsymbol{s}] = \mathbb{E}[\boldsymbol{\beta}_* + \boldsymbol{L}\boldsymbol{\eta}] = \boldsymbol{\beta}_* + \boldsymbol{L}\mathbb{E}[\boldsymbol{\eta}] = \boldsymbol{\beta}_*$, and secondly,

  $$
  \begin{aligned}
  \boldsymbol{\Gamma}_s &= \mathbb{E}[(\boldsymbol{s} - \boldsymbol{\beta}_*)(\boldsymbol{s} - \boldsymbol{\beta}_*)^T] \\
  &= \mathbb{E}[(\boldsymbol{\beta}_* + \boldsymbol{L}\boldsymbol{\eta} - \boldsymbol{\beta}_*)(\boldsymbol{\beta}_* + \boldsymbol{L}\boldsymbol{\eta} - \boldsymbol{\beta}_*)^T] \\
  &= \mathbb{E}[(\boldsymbol{L}\boldsymbol{\eta})(\boldsymbol{L}\boldsymbol{\eta})^T] \\
  &= \boldsymbol{L}\mathbb{E}[\boldsymbol{\eta}\boldsymbol{\eta}^T]\boldsymbol{L}^T \\
  &= \boldsymbol{L}\boldsymbol{L}^T \\
  &= \boldsymbol{\Gamma}_\beta
  \end{aligned}
  $$

  In the manuscript, for computational efficiency, we carry this process out using the procedure given in [9, Equation (30)].

- **Table 1:** Report units of $\delta$ and $\gamma$. Also, in my understanding $\sqrt{\frac{\gamma}{\delta}}$ is a correlation length for the samples. It might be worth pointing that out.

  We have specified the units of $\delta$ (adimensional) and $\gamma$ (m$^2$) in the manuscript, and have reported the corresponding correlation lengths (which is indeed proportional to $\sqrt{\gamma/\delta}$) for each of the examples in the caption of Table 1.

- **sect. 5.1:** In each of the 3 examples, I would remind the reader that $a_{\text{true}}$ is chosen as previously shown in Figures 3 and 4.

  Done.

- **p. 25, l. 17:** I would rephrase this sentence "The rheology parameters of the ice, in particular the flow rate factor and the Glen's flow law exponent, were treated as unknown random fields, which is often the case in reality." In real application the rheology is computed out of the temperature based on physical models.

We agree that some rheological parameters, such as the flow rate factor, are often computed using physical models, such as the Arrhenius relation, as in [6]. The temperature is, however, itself uncertain, as it is typically only measured on the surface of the ice. Thus, the flow rate factor is still uncertain since it depends on the unknown (or uncertain) temperature field. On the other hand, we are not aware of physical models for other rheological parameters, such as the Glen's flow law exponent, which are often fixed to nominal and approximate *best-fit* values. We have modified the sentence in an attempt to make this point clearer.

**References**

[1] T. Isaac, N. Petra, G. Stadler, and O. Ghattas, *Scalable and efficient algorithms for the propagation of uncertainty from data through inference to prediction for large-scale problems, with application to flow of the Antarctic ice sheet*, Journal of Computational Physics, 296 (2015), pp. 348–368.

[2] T. M. Kyrke-Smith, G. H. Gudmundsson, and P. E. Farrell, *Can seismic observations of bed conditions on ice streams help constrain parameters in ice flow models?*, Journal of Geophysical Research: Earth Surface, 122 (2017), pp. 2269–2282.

[3] K. Muhumuza, J. M. Huttunen, L. Roininen, and T. Lahivaara, *A Bayesian-based approach to improving acoustic Born waveform inversion of seismic data for viscoelastic media*, Inverse Problems, (2020).

[4] M. Perego, S. Price, and G. Stadler, *Optimal initial conditions for coupling ice sheet models to Earth system models*, Journal of Geophysical Research: Earth Surface, 119 (2014), pp. 1894–1917.

[5] N. Petra, J. Martin, G. Stadler, and O. Ghattas, *A computational framework for infinite-dimensional Bayesian inverse problems: Part II. Stochastic Newton MCMC with application to ice sheet inverse problems*, SIAM Journal on Scientific Computing, 36 (2014), pp. A1525–A1555.

[6] N. Petra, H. Zhu, G. Stadler, T. J. R. Hughes, and O. Ghattas, *An inexact Gauss-Newton method for inversion of basal sliding and rheology parameters in a nonlinear Stokes ice sheet model*, Journal of Glaciology, 58 (2012), pp. 889–903.

[7] M. R. Pralong and G. H. Gudmundsson, *Bayesian estimation of basal conditions on Rutford Ice Stream, West Antarctica, from surface data*, Journal of Glaciology, 57 (2011), pp. 315–324.

[8] M. J. Raymond and H. Gudmundsson, *Estimating basal properties of ice streams from surface measurements: a non-linear Bayesian inverse approach applied to synthetic data*, The Cryosphere, 3 (2009), pp. 265–278.

[9] U. Villa, N. Petra, and O. Ghattas, *hIPPYlib: An Extensible Software Framework for Large-Scale Inverse Problems Governed by PDEs; Part I: Deterministic Inversion and Linearized Bayesian Inference*, arXiv e-prints, (2020). In review.

---

## Referee Report (RR1)

The authors did only minor edits to the paper. I would encourage the authors to incorporate into the paper most of the arguments and numerical results used when replying to the reviewers. I think that results similar to the ones reported in the two Figures of their responses to the reviewers, and the related discussion, should be incorporated into the paper, as they address relevant issues. In general, I think that most of my concerns required to be addressed in the paper, not only in the response to the referee. As an example, the motivation of the Gaussianity assumption and its link to Tikhonov regularization provided in the response would be useful to the reader of this journal.

I am fine with most of their replies to my questions. Here is a point that I would like to further discuss:

1. ORIGINAL COMMENT: While I am convinced of the effectiveness and usefulness of the proposed approach, I am wondering whether the difference between the proposed approach and the traditional one has been overemphasized by taking a regularization (prior) for $\beta$ that is too small. In fact, it seems to me that the data are overfitted when using the traditional approach (REF/CEM). It would be interesting to see what happens when $\gamma$ and $\beta$ are increased (one could do that using the L-curve rule, for the deterministic inversion to compute the MAP point). In general, I think that the parameters used for all the priors should be motivated.

   AUTHORS RESPONSE: It would be extremely difficult (likely impossible), to take into account the correlation structure embedded in (the covariance matrix of) the approximation errors via a regularization parameter, thus loosing valuable information. Furthermore, the mean of the approximation errors is not negligible in all cases, that is, disregarding the unknown rheological parameters induces systematic bias, which would be challenging (at best) to accommodate by changing $\gamma$ and $\beta$.

   NEW COMMENT: I was not arguing that it is possible to account for the correlation structure in the approximation errors via a regularization parameter, nor that changing $\gamma$ and $\beta$ could prevent from systematic bias arising from disregarding unknown parameters. I was saying that it might be more fair, or at least informative, to increase the regularization for the traditional approach (REF/CEM) to avoid overfitting the data. For this reason I would recommend that the authors added a case where the regularization is significantly increased (e.g. scaling the prior covariance by a factor of 10 or 100), so that the data are not over-fitted. This might make the paper conclusion stronger, if it turns out that the amount of regularization required to avoid overfitting would make the results hardly useful.

---

## Author Response (AR2)

**Reply to the Reviewer:**

Thanks for the careful reading and for your helpful comments and suggestions for our revised manuscript. Please find below point-by-point replies (in black) to your comments (which are reprinted in blue). To give you an overview of all the changes in the updated manuscript, we also provide a diff-document that highlights the changes between the initial submission and this re-submission.

The authors did only minor edits to the paper. I would encourage the authors to incorporate into the paper most of the arguments and numerical results used when replying to the reviewers. I think that results similar to the ones reported in the two Figures of their responses to the reviewers, and the related discussion, should be incorporated into the paper, as they address relevant issues. In general, I think that most of my concerns required to be addressed in the paper, not only in the response to the referee. As an example, the motivation of the Gaussianity assumption and its link to Tikhonov regularization provided in the response would be useful to the reader of this journal. I am fine with most of their replies to my questions. Here is a point that I would like to further discuss: ORIGINAL COMMENT: While I am convinced of the effectiveness and usefulness of the proposed approach, I am wondering whether the difference between the proposed approach and the traditional one has been overemphasized by taking a regularization (prior) for $\beta$ that is too small. In fact, it seems to me that the data are overfitted when using the traditional approach (REF/CEM). It would be interesting to see what happens when $\gamma$ and $\beta$ are increased (one could do that using the L-curve rule, for the deterministic inversion to compute the MAP point). In general, I think that the parameters used for all the priors should be motivated. AUTHORS RESPONSE: It would be extremely difficult (likely impossible), to take into account the correlation structure embedded in (the covariance matrix of) the approximation errors via a regularization parameter, thus loosing valuable information. Furthermore, the mean of the approximation errors is not negligible in all cases, that is, disregarding the unknown rheological parameters induces systematic bias, which would be challenging (at best) to accommodate by changing $\gamma$ and $\beta$. NEW COMMENT: I was not arguing that it is possible to account for the correlation structure in the approximation errors via a regularization parameter, nor that changing $\gamma$ and $\beta$ could prevent from systematic bias arising from disregarding unknown parameters. I was saying that it might be more fair, or at least informative, to increase the regularization for the traditional approach (REF/CEM) to avoid overfitting the data. For this reason I would recommend that the authors added a case where the regularization is significantly increased (e.g. scaling the prior covariance by a factor of 10 or 100), so that the data are not over-fitted. This might make the paper conclusion stronger, if it turns out that the amount of regularization required to avoid overfitting would make the results hardly useful.

Thank you for the clarification and recommendation. We have added a Supplementary Material to the manuscript, where we incorporated several numerical examples recommended by the reviewers. In the revised manuscript, we refer to these new studies in the Numerical Examples section (see Section 5). Specifically, via these new numerical examples we study the effect of

- using a true auxiliary parameter, $a_{\text{true}}$, which is a given function rather than a sample from the associated prior,

- using different (in terms of mean and variance) prior distributions for the basal sliding coefficient, and

- using a reguralization-type approach, where we tune/scale the parameters of the prior (as suggested above) and likelihood, to account for the approximation errors.

---

## Author Response (AR3)

NOEMI PETRA

School of Natural Sciences
University of California, Merced
5200 N. Lake Road
Merced, CA 95343
email: npetra@ucmerced.edu

Prof. Gagliardini
Associate Editor
The Cryosphere

February 8, 2021

Dear Prof. Gagliardini:

We are pleased to submit the revised manuscript "Inferring the basal sliding coefficient for the Stokes ice sheet model under rheological uncertainty" by Olalekan Babaniyi, Ruanui Nicholson, Umberto Villa, and Noemi Petra for publication in The Cryosphere. In the revisied manuscript we have added a new subsection (please see Subsection 6.4 in the attached diff file or in the manuscript) that outlines the additional examples we added to the Supplementary Material as well as summarised our findings for the three new test cases.

We thank again the reviewers for the thorough comments, and many thanks to you for handling the manuscript with such care.

Sincerely,

Noemi Petra